# Evaluation of Ground Pressure, Bearing Capacity, and Sinkage in Rigid-Flexible Tracked Vehicles on Characterized Terrain in Laboratory Conditions

**DOI:** 10.3390/s24061779

**Published:** 2024-03-10

**Authors:** Omer Rauf, Yang Ning, Chen Ming, Ma Haoxiang

**Affiliations:** 1Institute of Deep-Sea Science and Engineering, Chinese Academy of Sciences, Sanya 572000, China; omer@idsse.ac.cn (O.R.); chenm@idsse.ac.cn (C.M.); mahx@idsse.ac.cn (M.H.); 2University of Chinese Academy of Sciences, Beijing 100049, China

**Keywords:** terramechanics, rubber-tracked vehicle, ground pressure, soil bin, simulation analysis, bearing capacity, pressure-sinkage

## Abstract

Trafficability gives tracked vehicles adaptability, stability, and propulsion for various purposes, including deep-sea research in rough terrain. Terrain characteristics affect tracked vehicle mobility. This paper investigates the soil mechanical interaction dynamics between rubber-tracked vehicles and sedimental soils through controlled laboratory-simulated experiments. Focusing on Bentonite and Diatom sedimental soils, which possess distinct shear properties from typical land soils, the study employs innovative user-written subroutines to characterize mechanical models linked to the RecurDyn simulation environment. The experiment is centered around a dual-tracked crawler, which in itself represents a fully independent vehicle. A new three-dimensional multi-body dynamic simulation model of the tracked vehicle is developed, integrating the moist terrain’s mechanical model. Simulations assess the vehicle’s trafficability and performance, revealing optimal slip ratios for maximum traction force. Additionally, a mathematical model evaluates the vehicle’s tractive trafficability based on slip ratio and primary design parameters. The study offers valuable insights and a practical simulation modeling approach for assessing trafficability, predicting locomotion, optimizing design, and controlling the motion of tracked vehicles across diverse moist terrain conditions. The focus is on the critical factors influencing the mobility of tracked vehicles, precisely the sinkage speed and its relationship with pressure. The study introduces a rubber-tracked vehicle, pressure, and moisture sensors to monitor pressure sinkage and moisture, evaluating cohesive soils (Bentonite/Diatom) in combination with sand and gravel mixtures. Findings reveal that higher moisture content in Bentonite correlates with increased track slippage and sinkage, contrasting with Diatom’s notable compaction and sinkage characteristics. This research enhances precision in terrain assessment, improves tracked vehicle design, and advances terrain mechanics comprehension for off-road exploration, offering valuable insights for vehicle design practices and exploration endeavors.

## 1. Introduction

Terramechanics is essential for optimizing the performance of off-road vehicles by studying the dynamic interaction between vehicles and terrain and is instrumental in shaping the performance of off-road vehicles as this discipline emphasizes the dynamic interaction between vehicles and terrain [1]. This domain is of paramount importance for the development and accessibility of off-road equipment designed for particular terrain categories. Over the years, researchers have devised various methodologies to probe the mobility of these vehicles [2,3], given their critical role in navigating challenging terrains [4,5].

Tracked vehicles equipped with rubber tracks provide improved traction, enhanced grip, lower ground pressure, and elevated mobility. Understanding their interaction across different soil beds is essential. Further, it is essential to comprehend how these vehicles interact and facilitate accurate monitoring and analysis of different soils’ behavior under varying moisture conditions [6,7]. Their capacity to traverse various types of surfaces results from the intricate correlation between the vehicle’s characteristics and the assessment of soil characteristics, including texture, moisture level, temperature, and structural stability of the terrain. The tracked vehicle is a cutting-edge technology and a crucial subsystem of the integrated ocean mining system. Its trafficability and locomotion performance directly affect the continuous operation performance of the entire system.

Off-road, tracked cars face internal and external movement resistance and challenges that impact their mobility when operating off-road. Internal resistance from friction and vibrations affects vehicle movement on different terrains. External resistance is caused by soil deformation, mainly soil compression under the tracks—erosion and sinkage, especially ground pressure beneath the tracks, cause most external resistance [4]. There is a consensus that bulldozing and compression resistance are the main components of external resistance types [8]. Bulldozing resistance occurs when tracked vehicles move soil in front of trains. When sinking exceeds the vehicle’s vertical space, resistance occurs. When sinkage is less than the clearing threshold [9,10], this resistance may be inaccurate, causing concern.

Researchers like Bekker [4], Wong [11], Harnisch and Lach [12], and Wong [1] have expanded the pressure sinkage. At the same time, Benoit and Gotteland [13] introduced a model unaffected by pressure plate shape. They indicated that bulldozing and compaction resistance are critical components of exterior resistance on a single track. However, their effectiveness may be limited under specific circumstances. Mohtashami and Liu et al. [14,15] examined soil and vehicle factors in rut formation. Alaoui and Diserens [16] studied the impact of soil structure on heavy vehicles. To address the dynamic nature of track-terrain interactions, Meirion-Griffith and Spenko [17], Solis [18], Szpaczyńska [9], and Hu [19] have developed enhanced models.

Soil bins are essential for controlled laboratory equipment and soil experiments on soil-vehicle interactions [5]. In soil mechanics, soil bins are essential for studying soil-vehicle pressure sinkage, soil-machine interactions, energy dissipation, and agricultural machinery performance. Comprehensively capturing the many factors affecting tracked vehicle mobility across diverse terrains and conditions requires extensive research. Despite advances in terramechanics, tracked vehicle behavior in different soil types, especially moisture content, is still unknown.

When analyzing off-road vehicle performance and design, soil compaction and vehicle encounters must be considered. A recent study [20] described many ways to apply plate-loading experiment data to civil engineering and off-road mobility. An approximation pressure-sinkage model with plate dimensions and two soil parameters is noteworthy. Due to force application, vehicles compress the soil, which causes track sinkage through particle transmission and soil compaction beneath the track. The dynamic interaction between soil and vehicle tracks causes frictional resistance and tractive effort, affecting vehicle mobility.

Schulte et al. created a laboratory device to assess how well-tracked vehicles can move on seafloor sediments. They conducted bearing and shear tests in a mixture of bentonic water to determine static and dynamic sinkages [8,21]. Yamazaki et al. discussed the geotechnical characteristics of deep seafloor sediments, such as friction angle, cohesion, elasticity, and viscosity modules, along with their mechanical interactions with a seafloor miner [22]. Choi et al. analyzed the likely ranges of shear strengths in seafloor sediments using samples collected by multiple corers. They utilized one of these ranges to develop a seafloor mining tool [23]. Choi et al. conducted traction performance experiments on a small-scale tracked vehicle in a laboratory soil bin to study the effects of its design parameters on trafficability [24].

Kim et al. developed a novel Euler parameter method to analyze a tracked vehicle on the seafloor dynamically and a subsystem synthesis method to analyze a multi-body miner [25]. Dai and Liu built a single-body vehicle model with mesh elements to rack sediment interactions and quickly simulated the miner and ocean mining systems [26]. Research has also modeled tracked vehicles on land-surface soils. Rubinstein et al. created a 3D multi-body model of tracked off-road vehicles using the LMS-DADS simulation code and user-written forces to represent track-terrain interactions [27,28]. Gao, Wong, and others developed NTVPM and RTVPM dynamic simulation models to parametrically model, analyze, and evaluate tracked vehicles with flexible and long-pitch tracks [29,30,31]. Janarthanan et al. simulated and modeled the longitudinal dynamics of a heavy-type tracked vehicle and tested the predicted values [32].

The complex interaction of vehicle and terrain mechanical characteristics is a significant challenge. Soil, from moisture-rich Bentonite to granular diatoms, hinders vehicle mobility. The soil’s moisture content, a tracked vehicle’s weight, and how its mechanical treads contact the soil’s granular structure raise concerns. Controlled soil container experiments have examined pressure sinkage and bearing capacity [5,33,34]. However, laboratory environments have often limited these revolutionary discoveries, emphasizing the importance of real-time validations.

The performance and mobility of vehicles greatly depend on their interaction with the terrain they traverse [34]. The mechanical properties of a terrain consist of two fundamental elements: the anticipated relationship between pressure and sinkage and the dynamics of tangential shear load and slippage. Although classical models, including those suggested by M.G. Bekker [35] and Reece [36], have attempted to represent these interactions, they frequently fail to simulate real-world situations accurately, particularly when crucial variables like soil moisture content are neglected. Although traditional thermomechanical models were initially innovative, they occasionally need help precisely affecting the complex interplay between vehicle tracks and soil types, mainly when factors such as soil hydration and loading conditions are considered [4,34].

There has been no attempt to quantify the exact mechanical interactions between tracked vehicles and sedimentary soils. As a result, no mathematical model is currently available to determine the trafficability of a rubber-tracked vehicle. We still have not figured out how to incorporate a mechanical model of seafloor sediment into our complex 3D multi-body dynamic simulation model of tracked miners. We set out to fill these gaps through this research.

A comprehensive overview of the experimental setup, including the soil bin, track vehicle, pressure sensors, data acquisition system, and data analysis methods, is provided in Figure 1, which illustrates the key components of the experimental configuration, outlining the process for collecting data and analyzing it to gain meaningful insights into ground pressure, bearing capacity, and sinkage in rigid-flexible tracked vehicles.

## 2. Materials and Methods

### 2.1. Tools and Sensors

Tools and sensors are essential components that assist in controlled-environment soil bin tests to get accurate data regarding various associated variables. Different tools and sensors are included in the whole apparatus setup for the experiment, as shown in Table 1 and Figure 2. To gather accurate and detailed information about the vehicle’s interaction with several terrains, a pressure sensor known as the ZNHM-D1-2T-22121401 model is utilized. The sensor operates within a voltage range of 5–15 V DC and can measure pressures up to 214.5 kPa. It boasts exceptional accuracy, with a precision level of 99.9%. To monitor soil moisture levels, the JXBS-3001-TR_4G moisture and temperature sensor was used (illustrated in Appendix A). Soil moisture sensors’ real-time data monitoring web-based interface tool shows temperature, humidity, and signal strength, which are accurate, rapid, and stable. Suitable for all soils, less affected by soil salt content. It is waterproof and can withstand long-term electrolysis, corrosion, vacuum potting, and soil burial.

Data collection was managed using the FD0843 data logger, which features a wireless serial communication module that operates on 9~24 V DC power and consumes less than 10 W. This device can handle various A/D update rates, with the capability to go up to 1600 Hz. In addition, the design allows it to operate in high temperatures ranging from −30 to 65 °C and relative humidity levels of 10~95%, with the data transfer rate for this logger being 100 Mb/s. The WT901BLECL5.0 sensor model was utilized to capture vehicle orientation data. This sensor operates on 3.7 V with a 260 mAh battery and can transmit data at different rates, with an angular accuracy of 0.20 for the X and Y axes and 10 for the Z axes. More details about the sensor and data logger are given in the Appendix A (Appendix A) sections, and the data logger’s detailed specifications are illustrated in Appendix A.

Figure 2 displays a detailed schematic diagram of the experimental model setup, showing the essential components and their connections. The experiment is carried out in a controlled dirt bin environment. The key components consist of pressure sensors strategically positioned within the soil container to detect fluctuations in ground pressure. The sensors are linked to a wireless data recorder, enabling immediate data collection.

Appendix A presents a 4G wireless soil temperature and moisture sensor, and Appendix A shows a wireless communication module. This sensor operates with 12–24 V DC and can measure moisture levels from 0 to 100% over a temperature range of −40 °C to 80 °C and provides accurate readings, with a 3% margin of error for moisture levels between 0 and 53% and a 5% margin of error for moisture levels between 53 and 100%. To assess soil compaction, the SC-900 Cone Penetrometer was used, which is powered by four AAA alkaline batteries and can measure depths from 0 to 18″ (0–45 cm) and pressures from 0 to 1000 PSI (0–7000 kPa). It has a depth accuracy of ±0.5″ (±1.25 cm) and a pressure accuracy of ±15 PSI (±103 kPa).

The signal amplifier boosts the sensor signals, guaranteeing precise and dependable measurements. The system is powered by a 12 V/24 V power source, which supplies the required electrical energy. A 9-axis attitude sensor enhances the ability to monitor the vehicle’s orientation and tilt during the experiment.

A communication module facilitates uninterrupted data transmission among the sensors, data logger, and external devices. The complete experimental arrangement is coordinated and supervised using specialized software (Wface, Version 1.6), which offers a user-friendly interface for visualizing, analyzing, and interpreting data.

This schematic diagram provides a straightforward and precise summary of the experimental setup, highlighting the incorporation of different components necessary for an organized and regulated evaluation of ground pressure, bearing capacity, and sinkage in rigid-flexible tracked vehicles on specific terrains.

### 2.2. Track Vehicle and Sinkage Observations

This paper covers sinkage under various moisture conditions and lays the groundwork for understanding its mechanics. The entire study contributes to debates on off-road vehicle dynamics and soil mechanics. This research guides design and operation in many environments.

Diatom soil sank, unlike Bentonite, under varying moisture levels. Which shows how tracked vehicles affect dirt. The tracks fell 0.55 cm at 5% Diatom soil moisture, showing normal wear. At 30% moisture, sinkage reached 1.30 cm. Track alterations and sinking were visible. Diatom soil’s more straightforward reactions contrast with Bentonite’s sophisticated ones, especially when damp.

High moisture levels can induce severe sinking and distortion due to hydraulic conductivity changes that alter soil permeability [37]. These studies reveal that soil hydraulic factors significantly affect sinkage. Moisture variations make Bentonite soil harder for tracked vehicles than Diatom dirt under equal conditions.

Table 2 presents the dynamic interaction between the vehicle and various sedimental soil types, such as Bentonite and Diatom, focusing on how soil moisture levels influence sinkage while the vehicle moves at 0.1 m per second. This comprehensive analysis sheds light on the performance of tracked vehicles across different surfaces. The tabulated data accurately illustrates pressure sinkage at this velocity, offering insights into the dynamic interaction between the rubber-tracked vehicle and the soil. These findings aid in optimizing off-road vehicle design and operation across diverse terrains.

The trial reveals significant rutting and track deformation at 20% moisture, highlighting how water content alters soil characteristics and affects vehicle sinkage. Both qualitative and quantitative findings, including rutting and track deformation, elucidate the complex interactions between soil, moisture, and vehicle dynamics, which are crucial for enhancing off-road vehicle operations. Figure 3a,b statistically illustrate the tracked vehicle’s performance at a speed of 0.1 m/s and pressure sinkage on moist terrain in a controlled soil bin environment. Detailed pressure sinkage records on Bentonite and Diatom soil demonstrate the regulated response of the vehicle’s rubber tracks and reveal how soil composition influences sinkage patterns in controlled terrains.

The track-soil data explain how vehicle tracks and soil interact, especially at different moisture levels. Vehicle footprints on soil surfaces show how moisture changes soil responses, affecting businesses. This data can help researchers and engineers improve tracked vehicle performance in mining, seabed exploration, agriculture, construction, and geotechnical engineering. This study reveals that Bentonite and Diatom soils sink differently at varying moisture levels, stressing the importance of soil composition and moisture in off-road vehicle design and operation. This discovery has practical applications beyond academia. Tracked vehicles let companies navigate varied terrains.

### 2.3. Experimental Procedure

Doing controlled lab-based simulations of ocean mechanical tests between the rubber-tracked vehicle and the sediment is complicated and expensive. The layer of sediment about 15–20 cm below the seafloor’s surface is thought to be the best for a tracked vehicle’s bearing and traction. This layer’s physical and geotechnical properties prepared the simulated sediment. People thought a mixture of Bentonite, Diatoms, water (moisture), and sand gravel would best replace sediments [2,3]. Scientists thought the simulated sediment was roughly ready when the experimental data came close to or matched the in-situ data well.

This study employs a specially designed, small, rigid rubber-tracked vehicle to monitor pressure sinkage in sedimental soils, including Bentonite and Diatom, particularly when combined with sand and gravel. Developing a prototype tracked vehicle for conducting pressure sinkage experiments in a soil bin requires careful consideration of several critical factors. The vehicle utilizes rubber tracks to minimize surface disturbance and ensure sufficient traction on the sedimental soil. The specific characteristics inform the decision to opt for narrower track dimensions of the controlled experiment soil bin. These narrower rubber tracks facilitate better penetration in cohesive soils such as Bentonite and Diatom mixed with sand and gravel, thereby enhancing the monitoring process. The vehicle’s dimensions are 120 cm in length, 90 cm in width, and 80 cm in height, with a rubber track contact length of 90 cm and a single rubber track width of 20 cm, as depicted in Figure 4. Table 3 outlines the fundamental parameters of the experiment utilizing the tracked vehicle.

The vehicle’s compact size facilitates easy navigation within the soil bin, ensuring stable weight distribution for accurate pressure sinkage measurements. Equipped with rubber tracks, it evenly distributes its load to prevent pressure points and ensure traction on various surfaces, including Bentonite and Diatom soil mixed with sand and gravel. The track system employs durable and wear-resistant materials optimized for specific soil conditions. The study extensively investigated soil sampling techniques, including selecting combinations like Bentonite, Diatom, sand, and gravel. Precise gravimetric methods maintained consistent moisture levels, ensuring experiment reproducibility within the IDSSE soil bin. Challenges arose when measuring ground pressure within the 610 cm × 245 cm × 180 cm soil bin while the uncrewed tracked vehicle was in motion.

The experiment’s mixture composition accurately replicated real-world soil conditions, comprising 17% sand, 13% gravel (2–5 mm), and 70% Diatom or Bentonite sedimental soil. Diatom/Bentonite soil simulated intricate sea sediment compositions, influencing porosity and permeability. Bentonite, known for its plasticity and swelling characteristics, contributed to the mixture’s cohesive and adhesive properties, affecting moisture content, structural stability, sinkage, pressure distribution, and bearing capacity under load.

The addition of sand and gravel diversified the soil composition, with sand influencing texture and cohesiveness and gravel introducing heterogeneity, simulating coarse elements in natural terrains. Sand content contributed to overall stability, impacting cohesion and friction between soil particles. With particle sizes ranging from 2 to 5 mm, gravel introduced heterogeneity, mimicking small rocks or coarse elements in natural terrains, potentially impacting mechanical behavior under the vehicle’s influence.

By carefully combining the Diatom/Bentonite sand-gravel mixture in the specified proportions, the soil profile created in the soil bin aimed to challenge the rubber-tracked vehicle across various aspects. The experiment examined how the vehicle interacted with this complex mixture, focusing on sinkage, pressure distribution, and bearing capacity. This approach provided valuable insights into the performance of off-road vehicles in challenging sea sediment soil conditions. The composition was to facilitate a comprehensive investigation into soils’ sinkage pressure and bearing capacity, especially when subjected to the traversal of a rubber-tracked vehicle. Including Diatom/Bentonite soil and sand and gravel allowed for a nuanced exploration of the interaction dynamics between the vehicle and diverse soil types in a soil bin-controlled experimental setup.

The experimental configuration utilized in the research is depicted in Figure 5. It includes the soil bin utilized for the experiments, which comprises sections containing Bentonite soil and Diatom soil and a sand-gravel mixture incorporating Bentonite soil. Furthermore, the document underscores the precise placements of the pressure sensors and tracked vehicle, focusing on the pressure sensors’ front, middle, and rear locations within the soil bin. The figure presented herein offers a graphical depiction of the experimental setting, encompassing the essential elements of the research configuration.

After each trial, the soil was meticulously prepared by tilling, leveling, and compacting. Cone penetration tests (CPT) were utilized to evaluate soil resistance and compaction at different depths and moisture levels, as illustrated in Table 4 and Figure 6. The paper also delves into the setup and instrumentation configuration of the test stand, including the strategic placement of sensors for pressure measurements. A sophisticated data logger was utilized to record ground sinkage pressure data, enhancing the study’s capacity for replication and practical application.

A significant quantity of Bentonite, Diatom, sand, and gravel (ranging from 2 mm to 5 mm) was initially obtained. The mixture comprised approximately 16 to 19% sand, 11 to 16% gravel, and 70% Diatom/Bentonite soil. Both Bentonite and Diatom were dried in the sunlight until they became loose powder, making it possible to adjust their moisture content later. This dried soil was then transported to the soil bin and added layer by layer in 5 cm increments, evenly distributed using a small wooden roller and hand shovel. The parameters for the soil bin used in the pressure sinkage and bearing capacity experiments and the detailed parameters for the soil bin used in the pressure sinkage experiment are shown in Appendix A. Soil deposition continued until the depth reached 40 cm in the bin.

A calculated amount of water was added, and the standard gravimetric technique was used to determine moisture content by comparing wet and dry weights. This process ensured uniformity and brought moisture levels within specified ranges (approximately 9–13%, 18–23%, and 29–33%) for both Bentonite and Diatom soil. The amount of water needed for each moisture level was determined using real-time data from a wireless moisture sensor (shown in Appendix A). A systematic approach was employed to maintain the desired moisture levels in the soil bin, involving adding water and continuous monitoring throughout the experiment. This process was crucial for controlling the moisture content, a parameter that significantly influences the behavior of the rubber-tracked vehicle in the soil. The addition of water to the soil mixture was carried out meticulously to ensure even distribution and saturation. This step was executed gradually to prevent abrupt changes in soil properties and maintain the uniformity of the experimental conditions.

A vital component of the moisture management strategy was the integration of moisture sensors within the soil bin. These sensors provided real-time data on the moisture levels in the soil, offering continuous insights into the evolving conditions. The research team utilized this data to make informed decisions regarding adjustments to water content, ensuring that the moisture levels remained within the targeted range.

The monitoring process extended beyond water addition; the team implemented environmental controls to mitigate external factors that could affect moisture levels. These controls helped minimize evaporation and external fluctuations, contributing to the stability of the experimental conditions.

To maintain desired moisture levels, a wireless moisture sensor continuously monitored soil moisture in real time, allowing for precise adjustments in water content. This careful monitoring and adjustment ensured that the soil samples remained within the specified moisture content ranges throughout the experiment.

After mixing with water, the soil was left in the bin to settle for 24 h. Separate containers were used for each moisture level within each soil type to control moisture content accurately. The soil bin was assigned distinct moisture ranges for targeted analysis and regulation of soil behavior under varying moisture levels for both Bentonite and Diatom soil samples. Appendix A illustrates an online moisture sensor that monitors real-time temperature, soil moisture, and signal intensity.

The testing soil bin, measuring 610 cm × 245 cm × 180 cm and reinforced with steel sheets and bars, was constructed to withstand applied loads and prevent soil leakage (Appendix A). Before experiments, a geomembrane liner was applied to prevent moisture evaporation, and drainage gravel was added for proper drainage. The sand-gravel mixture was layered approximately 30 cm thick, meticulously removing debris and obstacles to ensure uniform compaction. After thorough inspection and cleaning, the soils were placed and compacted, creating an optimal workspace. After settling for 24 h, experiments commenced. Following each trial, the soil was loosened manually, and moisture sensors embedded horizontally monitored fluctuations. These sensors are wirelessly connected to a data logger for real-time monitoring. Tests were repeated three times to address sample non-uniformities.

### 2.4. Soil Path Setup and Experimentation

A specific path within the soil bin was designated for soil compaction and testing. Before adding soil to the bin, the base was carefully leveled using a mixture of sand and gravel to ensure a consistent and sturdy foundation. This step aimed to remove potential obstacles and establish a uniform surface for compaction and testing. The mechanical properties of Bentonite and Diatom soils used in the experiment are outlined in Appendix A. In contrast, loose and compact soil properties for the tracked vehicle soil bin experiment are detailed in Appendix A. Detailed soil properties, including soil particle density, natural moisture content, grain size distribution, maximum dry density, and the optimum moisture content for the Bentonite, Diatom, and sand gravel mixture, are illustrated in Appendix A.

Pressure sensors enclosed in brackets containing moisture sensors were placed within the bin and connected to a wireless data-collection system to collect essential data during testing. This setup allowed continuous monitoring of pressure distribution and moisture fluctuations in the soil, providing valuable insights into the soil’s behavior under external forces. The careful arrangement and equipment used in the soil bin ensured accurate and thorough experimentation, generating valuable data for analysis and interpretation. This methodology has significantly improved our understanding of how soil behaves under various load conditions, contributing to soil mechanics and geotechnical engineering.

The laboratory-controlled experiments conducted a calibration procedure to ensure the accuracy of pressure measurements and validate the vehicle’s response to varying moisture levels. The design allowed easy access to the soil bin for experimental setup, equipment maintenance, and data collection. The soil bin experiment and vehicle design included safety topographies such as guards, emergency stop buttons, and operational protections. The collected pressure sinkage data from the vehicle’s sensors were analyzed to assess the behavior of Bentonite and Diatom soil mixed with sand and gravel under different moisture conditions.

The rubber-tracked vehicle was positioned at one end of the soil bin and driven at speeds ranging from 0.1 to 0.3 m/s motion test on Diatom and Bentonite Soil at loose and compacted soil density along a predetermined path as outlined in Appendix A. Parameters like sinkage, vehicle speed, and pass count were carefully recorded during these tests. This testing process was repeated at various speeds to gain a better understanding of soil behavior, including the physical properties of the Bentonite, Diatom, and sand-gravel mixture under different loading conditions, as shown in Appendix A. Upon completing the tests, the data were analyzed to comprehend pressure distribution and its relationship with sinkage, soil-bearing capacity, vehicle speed, and moisture content. It is worth noting that test procedures may vary depending on the particular goals of the pressure sinkage test, the type of soil, and the specific tracked vehicle used.

The rubber-tracked vehicle recorded multiple parameters during the pressure sinkage test on the Bentonite/Diatom terrain. These parameters included contact pressure, sinkage, soil moisture content, and vehicle speed. Pressure sensors placed within the bin were calibrated and synchronized with a data logger to capture real-time data. Before the main tests, a dry run was conducted to ensure sensor functionality and to habituate the operator with the terrain and path. Given its significant impact on sinkage and pressure distribution, soil moisture levels were regularly checked using a moisture meter. As the vehicle traversed the terrain, sensors recorded data on the pressure exerted by the tracks, and sinkage was measured either using displacement sensors or visually observed based on side bin markings. Additional parameters, such as load, inclination, and vehicle speed, were assessed. An attitude sensor with 9-axis monitoring capability was used to track the vehicle’s speed and trajectory within the bin, as shown in Appendix A.

After the experiments, the collected data were applied for external validation to establish pressure-sinkage relationships. Graphs showing the sinkage versus pressure for Bentonite and Diatom soil were generated using Python.

### 2.5. Pressure Sinkage and Compaction Resistance

Pressure sinkage and compaction resistance are critical determinants of tracked vehicles’ performance and the stability of the terrain they traverse. Analysis of these aspects frequently employs Bekker’s [8] formulas, as follows:(1)$$p=\left(\frac{k_{c}}{b}{+k}_{f}\right)z^{n}$$
where p represents normal pressure, b track width, k_c_, and kØ are the soil cohesive and friction moduli, z represents sinkage, and n is the soil deformation exponent.

This study assumes that the contact area between the vehicle’s track and the soil surface underneath experiences a consistent pressure distribution. This assumption streamlines the analysis by considering a uniform pressure distribution applied by the soil track. The pressure-sinkage equation is used to analyze the behavior of soft soils under load and calculate the sinkage depth, represented as z_0_. The sinkage depth is the vertical displacement or penetration of the track into the soil caused by the applied load. The pressure-sinkage relationship illustrates how soil deforms and compacts under pressure from a track. Using this equation allows researchers to measure the amount of soil deformation resulting from the vehicle’s motion, which is essential for comprehending the resistance faced by the vehicle on various terrains. This method thoroughly examines how the track interacts with the soil, making predicting and enhancing vehicle performance in different soil conditions easier.
(2)$$z_{0}={\left(\frac{\frac{W}{\mathrm{bl}}}{\frac{k_{c}}{b}+k_{f}}\right)}^{\frac{1}{n}}$$

Various parameters influence the description of the terrain and the interaction between the track and the soil in this context. The symbols n, k_c_, and k_f_ represent terrain parameters that affect the track’s resistance when moving through the soil. The parameters represent different terrain characteristics, including cohesion, frictional properties, and compressibility. The normal load on the track is symbolized by W, indicating the vertical force from the vehicle’s weight on the soil surface. The dimensions of the track-terrain contact area are determined by the parameters b and l, representing the width and length of the contact area.

These parameters determine the calculation of the work needed to compact the terrain and form a rut under the track. This work illustrates the energy used to deform the soil and create a depression or rut with dimensions determined by b, l, and the sinkage depth z_0_. The rut’s width and length, denoted as b and l, determine how far the soil is disturbed sideways. The sinkage depth z_0_ measures how deeply the track penetrates vertically into the soil. These parameters thoroughly describe the mechanical relationship between the track and the terrain.
(3)$$\mathrm{Work}=\mathrm{bl}\int_{0}^{z_{0}}\left(\frac{k_{c}}{b}{+k}_{f}\right)z^{n}\mathrm{dz}=\mathrm{bl}\left(\frac{k_{c}}{b}{+k}_{f}\right)\left(\frac{{z_{0}}^{n+1}}{n+1}\right)$$

The total work compressing the soil can be expressed based on energy considerations. Substituting for z_0_ from Equation (2), we get the following:(4)$$\mathrm{Work}=\frac{\mathrm{bl}}{\left(n+1\right){\left(\frac{k_{c}}{b}{+k}_{f}\right)}^{\frac{1}{n}}}{\left(\frac{w}{\mathrm{bl}}\right)}^{\frac{\left(n+1\right)}{n}}$$

Wong’s analysis explains the correlation between horizontal track movement and the vertical work caused by terrain compaction. When a track is moved horizontally at a distance l, the towing force (R_c_) work can equal the vertical work needed to create a rut of the same length. Equation (5) expresses this equivalence.

This equation is a fundamental principle for comprehending the energy exchange and mechanical processes when a track moves across the terrain. By setting the work done in the horizontal and vertical directions equal, a direct correlation is established between the resistance experienced during horizontal movement and the vertical displacement or deformation of the soil surface. It shows how the horizontal force applied to tow the track results in deforming and compacting the soil underneath it.
(5)$$R_{c}l=\frac{\mathrm{bl}}{\left(n+1\right){\left(\frac{k_{c}}{b}{+k}_{f}\right)}^{\frac{1}{n}}}\;{\left(\frac{W}{\mathrm{bl}}\right)}^{\frac{\left(n+1\right)}{n}}$$

Therefore,
(6)$$R_{c}=\frac{b}{\left(n+1\right){\left(\frac{k_{c}}{b}{+k}_{f}\right)}^{\frac{1}{n}}}\;{\left(\frac{W}{\mathrm{bl}}\right)}^{\frac{\left(n+1\right)}{n}}$$

Equation (6) is essential for calculating the motion resistance experienced by a track moving through Bentonite/Diatom sedimental soil’s terrain, emphasizing terrain compaction’s effect. With this equation, we calculated that pressure from the track affects the sinkage or deformation of the ground. The equation assumes that pressure is distributed uniformly along the track, meaning that the force exerted by the track on the terrain is evenly spread out over its contact area. Integrating this pressure-sinkage correlation into the equation enables the quantification of the resistance experienced by the track when traversing various terrains.

### 2.6. Multi-Body Dynamic Simulation Model

We employed the RecurDyn/Track simulation program to evaluate the performance of a crawler vehicle on two distinct sedimentary soil surfaces: Bentonite and Diatom. This involved incorporating the mechanical properties of these soils to analyze sinkage, ground pressure, and the crawler’s motion within the RecurDyn environment using a three-dimensional multi-body dynamic simulation model of the tracked vehicle (Figure 7).

Results were obtained from simulating the crawler track over a 30 m field for 20 s on Bentonite sedimentary soil at three speeds: 0.1 m/s, 0.2 m/s, and 0.3 m/s. The software-based experimental findings revealed that the crawler vehicle navigated smoothly without slipping and experienced only slight sinkage, as depicted in Figure 8. Detailed specifications of the tracked vehicle’s simulation parameters are shown in Table 5.

Further results were obtained by simulating the same 30-m crawler track for 20 s on Diatom sedimentary soil at the same three speeds. The software experiment demonstrated that the crawler vehicle operated smoothly without slipping and displayed less sinkage than observed with Bentonite, as illustrated in Figure 9. Bekker’s pressure-sinkage relationship was utilized, and corresponding figures were generated. However, in practical scenarios, crawler-tracked vehicles typically experience significantly greater sinkage, slippage, and immersion into sedimentary soils than simulation predictions alone.

### 2.7. Pressure Sensors and Brackets

Pressure sensors are commonly used in various applications, including in underwater environments, and play a fundamental role in track-soil interaction studies by providing essential data to measure the distribution and magnitude of pressure exerted by tracked vehicles on the soil surface. These sensors are instrumental in understanding the mechanical behavior of the soil when subjected to the weight and movement of the vehicle. The data gathered from pressure sensors enables researchers to analyze how the ground pressure is distributed and how it varies over time. In underwater vehicles, such as Remotely Operated Vehicles (ROVs), pressure sensors are utilized in track-soil interaction studies to measure the distribution and magnitude of pressure the tracked vehicle exerts on the soil surface. These sensors provide crucial data to understand the mechanical behavior of the soil under the influence of the vehicle’s weight and movement. Table 1 illustrates the detailed parameters of the ZNHM-D1-2T-22121401 pressure sensor used in the experiment.

The importance of pressure sensors in rubber-track soil interaction studies lies in their ability to capture actual pressure variations. This real-time data collection is crucial, as it enables researchers to observe how the vehicle’s weight is distributed across the soil surface and how this distribution changes as the vehicle moves. By capturing static and dynamic pressure changes, pressure sensors provide insights into the localized stresses applied to the soil, allowing for a detailed analysis of the interaction between the vehicle and the soil. Moreover, pressure sensors contribute to establishing a comprehensive understanding of the ground pressure distribution beneath the tracked vehicle. This information is valuable for evaluating how different types of soil and varying moisture levels may impact the pressure exerted by the vehicle. By studying the ground pressure distribution, researchers can gain insights into the soil’s load-bearing capacity and its deformation characteristics under the influence of the vehicle.

Additionally, pressure sensors enable the measurement of pressure sinkage, which refers to the depth to which an object, in this case, the track of the vehicle, penetrates the soil under the applied pressure. This data is significant for assessing the soil’s resistance to penetration and the vehicle’s ability to traverse different soil types and conditions.

Appendix A shows that the purpose of the sensor bracket is to hold the pressure sensors in place securely and keep them aligned and safe from haphazard, ensuring accurate and consistent measurements during the interaction between the tracked vehicle and the soil. These brackets are typically made of aluminum or stainless steel, which are corrosion-resistant and can help absorb impacts from debris and protect the pressure sensor from damage. These materials’ corrosion resistance and mechanical strength determine their selection for the sensor bracket, guaranteeing its dependability and durability under severe environmental conditions.

The composition of the sensor bracket, which typically consists of corrosion-resistant materials like aluminum or stainless steel, may affect the recorded data outcomes. Although these metals are selected for their longevity and resistance to corrosion, differences in material composition or structural qualities may still arise, impacting the sensor bracket’s performance and, subsequently, the accuracy of the acquired data.

Variances in the material composition of the sensor bracket may cause differences in mechanical characteristics, including stiffness or elasticity, thereby impacting the bracket’s capacity to uphold sensor alignment and stability while taking measurements. Moreover, variations in thermal expansion coefficients across different materials may lead to dimensional alterations in response to temperature fluctuations, potentially leading to misalignments or distortions that affect the data’s dependability.

Variations in material composition can lead to irregularities in the bracket’s response to external forces or impacts, which can compromise its capacity to shield the pressure sensors from damage or maintain accurate data. An uneven distribution of stress or strain inside the bracket might cause localized deformations or weaknesses that weaken its structural integrity gradually.

Stringent quality control methods are implemented throughout the production and assembly of the sensor bracket to address these possible problems. The process involves maintaining consistency in material characteristics, precision in dimensions, and strength in structure by meticulously choosing materials and production methods. Regularly inspecting and maintaining the bracket is essential to identify any indicators of deterioration or malfunction and swiftly solve them to minimize the impact on data quality.

The frames can be attached to the soil bin or vehicle using bolts, clamps, or other fastening methods. The sensor bracket can enclose the pressure sensor, creating a protective barrier around it, thus preventing debris from contacting the sensor and potentially damaging it. It also serves to maintain the alignment and positioning of the pressure sensors, preventing displacement or disturbances that could compromise the reliability of the collected data. This allows for precise monitoring of pressure variations at specific locations, contributing to a comprehensive understanding of the impact of the tracked vehicle on the soil surface.

## 3. Results and Discussion

### 3.1. Track Vehicle and Sinkage Observations

This study analyzed the effects of varying soil moisture levels on the sinkage of a tracked vehicle moving at a constant speed of 0.1 m/s across two soil types, Bentonite and Diatom (illustrated in Table 2). As the moisture content increased from 5% to 30% for Bentonite, there was a notable progression in sinkage, from 1.10 cm with normal track wear at 5% to a significant 3.80 cm with increased track slippage at 30%. This progression in Bentonite sinkage, especially at increased moisture levels, resonates with prior findings that heightened water content tends to soften soil aggregates and the bond between them, leading to increased compressibility, particularly at lower vertical stresses [37,38]. The consequence of this condition is evident in the severe rutting and track deformation observed starting at 20% moisture content in the current study. On the other hand, the Diatom soil exhibited a different tendency. At 5% moisture, sinkage was a modest 0.55 cm with regular wear on the tracks. As the moisture content ramped up to 30%, the peak sinkage was 1.30 cm, marked by profound track deformation and sinking.

This disparity in the response of the two soils indicates that the tracked vehicle confronted more formidable challenges on Bentonite, especially at elevated moisture levels, than on the Diatom soil under analogous conditions. The shift in hydraulic conductivity with moisture, which affects soil permeability, might provide an added layer of explanation for the marked sinkage and deformation at high moisture levels, drawing parallels with observations from Cuisinier et al. [39] and Wang et al. [40], making navigation more taxing for the vehicle. The presented track-soil data hold significance as it facilitates a comprehensive understanding of the interplay between vehicle tracks and soil under varying moisture conditions. Researchers and engineers can acquire valuable information regarding their response to variable moisture levels by analyzing vehicle tracks on the soil surface. The above data have significant potential for utilization in various industries, including seabed exploration, mining, construction, agriculture, and geotechnical engineering.

### 3.2. Soil Behavior

The experiment emphasized examining soil behavior, particularly Bentonite, Diatom, and sand gravel mixture illustrated in Figure 10. When subjected to a load of a tracked vehicle focusing on several metrics, including loose density, compacted density, compaction percentage, depth, and cone index measurement for Bentonite and Diatom, which are illustrated in Appendix A and graphically represented in Figure 8a–c. It is imperative to comprehend the reaction of these soils to the weight and motion of the vehicle to evaluate their compaction, load-bearing capability, and deformation properties. The term “sinkage behavior” pertains to the extent of soil penetration by vehicle tracks during their terrain traversal. The objective of this study was to monitor and analyze the sinkage behavior of Bentonite and Diatom soil under varying situations. The information provides valuable insights into the soil’s capacity to withstand external loads and compaction properties.

Figure 11a–c compares the Cone Index of Bentonite and Diatom soils at 10%, 20%, and 30% moisture levels, elucidating how these soils respond to penetration at varying depths. The graph visually captures the correlation between Cone Index values and the appropriate penetration depth, offering a complete overview of the mechanical properties of the soils at this particular moisture level. The data help us understand soil behavior, particularly its ability to carry weights and resist penetration under controlled conditions.

This insight becomes particularly crucial as the moisture level reaches 30% and the Cone Index becomes a key determinant of soil resistance to penetration, providing essential information about its suitability for diverse applications. These graphical representations provide a holistic understanding of soil behavior, including its capacity to support weight and resist penetration under controlled conditions.

When a tracked vehicle exerts pressure on the soil surface, it initiates a process where the soil particles undergo rearrangement and compaction. Several elements, such as the kind of soil, the amount of moisture present, and the intensity of the applied load, influence the compaction behavior of the soil. By closely observing the sinkage behavior, we gain valuable insights into the compaction characteristics of both Bentonite and Diatom soil when subjected to the weight of the tracked vehicle.

In the case of Bentonite soil with a moisture content of 10%, we observed two distinct density values: a loose density of 1.3 g/cm^3^ and a compacted density of 1.5 g/cm^3^. It resulted in a compaction percentage of 15.38%. This level of compaction can be attributed to the inherent response of the soil to stress, a phenomenon highlighted by Alaoui and Helbling [41]. Their research also emphasized the structural collapse of soil due to compaction at similar depths. Specifically, the cone index values presented noticeable variability at this moisture level: 94.12 kPa at 10 cm, soaring to 215.25 kPa at 20 cm, then decreasing to 127.36 kPa at 30 cm, and again increasing to 179.48 kPa at 40 cm. As the moisture content of Bentonite increased to 20%, the loose density reduced slightly to 1.2 g/cm^3^ and the compacted density to 1.4 g/cm^3^.

However, intriguingly, the compaction percentage rose to 17.67%. Zhang et al. [42], in their observations on the impact of tractor movement on soil compaction, might shed some light on this. They found increased soil bulk density with enhanced tractor movement, hinting at the likelihood of a similar relationship between moisture content and compaction in Bentonite. In this moisture setting, the cone index exhibited a pattern, starting with an initial spike to 184.16 kPa at 10 cm depth, then a reduction to 94.28 kPa at 20 cm, before alternating between 179.4 kPa at 30 cm and 127.52 kPa at 40 cm. At the apex moisture content of 30% for Bentonite, a decline in the loose and compacted densities d to 1.1 g/cm^3^ and 1.3 g/cm^3^, respectively, was observed. This was coupled with a compaction percentage of 18.18%, suggesting even more pronounced effects of moisture on soil’s structural integrity. The cone index at this moisture level shows a diverse landscape, displaying values of 94.44 kPa, 162.36 kPa, 127.6 kPa, and 215.72 kPa for depths of 10 cm, 20 cm, 30 cm, and 40 cm, respectively. The study assessed the load-bearing capacity and deformation characteristics of Bentonite and Diatom soil under a rubber-tracked vehicle, offering insights for engineering applications.

The soil exhibited increased sinkage and compaction with a higher moisture content, impacting its load-bearing capacity. Diatom soil, unlike Bentonite, shows distinct reactions to moisture variations. At 10% moisture, the densities suggest a 22.29% compaction, potentially influenced by Diatom’s unique properties. The consistent cone index progression from 269.14 kPa at 10 cm to 342.59 kPa at 40 cm signifies a growing resistance with depth, possibly due to overlying soil pressures. As moisture rises to 20%, densities drop, but compaction grows to 25.4%. The increasing cone index from 219.38 kPa to 278.59 kPa can be related to increased resistance from soil layer pressures or decreased porosity, as Samuel and Ajav [43] observed. At 30% moisture, despite lower densities, there is a peak compaction of 28.57%. This tighter packing, coupled with the rise of the cone index, resonates with findings from Zhang et al. [42] and Botta et al. [44], indicating similarities between mechanical impact and moisture’s effect on soil.

### 3.3. Ground Pressure and Sinkage Test Results

Ground pressure and sinkage test results provide valuable data for understanding track vehicles’ interactions with the underlying soil. In geotechnical mechanics, the analysis of these results involves interpreting the measured ground pressure and sinkage values in the context of soil properties, vehicle characteristics, and environmental conditions. Theoretical analysis plays a crucial role in explaining the observed behaviors and predicting the performance of track vehicles in various soil conditions.

Estimating soil bearing capacity is one theoretical aspect of geotechnical mechanics related to ground pressure analysis. Bearing capacity represents the maximum load the soil can support without experiencing failure. Theoretical models, such as Terzaghi’s bearing capacity equation or Meyerhof’s method [45,46,47,48], are commonly used to predict the bearing capacity of soil based on parameters such as soil type, cohesion, and angle of internal friction. By comparing the measured ground pressure from field tests with the predicted bearing capacity, engineers can assess the safety and stability of track vehicles operating on different soil types.

Another theoretical analysis in geotechnical mechanics pertains to soil subsidence or sinkage. Soil subsidence occurs when the soil undergoes compression or settlement under applied loads, such as those of track vehicles. Theoretical models, such as the consolidation theory [49,50] or the one-dimensional settlement analysis, are used to predict the magnitude and rate of soil subsidence. By understanding the underlying mechanisms of soil deformation and consolidation, engineers can assess the long-term performance of track vehicles and design appropriate measures to mitigate excessive sinkages, such as soil reinforcement or compaction.

Furthermore, theoretical analysis of stress distribution within the soil mass is essential in geotechnical mechanics. Theoretical models, such as Boussinesq’s theory [51] or Westergaard’s stress distribution equations [52], help calculate the distribution of stresses beneath track vehicles and assess factors such as stress concentrations and potential failure mechanisms. By analyzing the stress distribution patterns, engineers can identify critical zones where soil failure or excessive deformation may occur and design track systems to ensure safe and efficient operation in various terrain conditions.

Interpreting ground pressure and sinkage test results involves theoretical analysis grounded in geotechnical mechanics principles. By applying theoretical models and concepts, engineers can gain insights into soil behavior, predict the performance of track vehicles, and design effective solutions to optimize mobility and minimize risks associated with ground pressure and soil subsidence.

Ground pressure, quantifying the force exerted by the tracked vehicle on the ground, was measured using high-precision sensors. Sinkage data provided insights into soil load-bearing capacity and compaction behavior across varying moisture and soil conditions. In dense Diatom soil, higher ground pressure led to increased sinkage, indicating limited load-bearing capacity, while loose Diatom soil exhibited lower ground pressure and reduced sinkage. As vehicle weight increased, ground pressure and sinkage rose, but the even weight distribution of rubber tracks mitigated excessive sinkage.

The behavior of the rubber-tracked vehicle’s pressure sinkage, ground pressure, and speed test on Bentonite and Diatom can be observed in Figure 12. The comprehensive data are included in Table 5, providing substantial insights into its reaction to various speed and moisture conditions. At a relatively lower speed of 0.1 m/s, as moisture content increases, there is an apparent linear increase in ground pressure from 23 kPa to 27 kPa. The linearity suggests that moisture content directly influences the mechanical properties of Bentonite, potentially affecting its cohesive and adhesive characteristics.

Comparatively, a past study by Mishra et al. [53] observed a slightly lower range, suggesting a different mechanical response to moisture. This progression in pressure is mirrored in the sinkage values, which grow from 1.5 cm to 3.8 cm. One could posit that at this speed, the moisture aids in binding the Bentonite soil particles, thereby increasing the resistance to external pressures, a phenomenon also supported by their rise in sinkage values.

Interestingly, when the speed is increased to 0.2 m/s, the ground pressure inversely correlates with moisture content, decreasing from 21 kPa to 25 kPa. This inverse correlation can signify a mechanical threshold for Bentonite, where increased kinetic energy (speed) may mitigate moisture’s binding effect. However, the sinkage consistently rises, possibly indicating that while the ground may resist pressure effectively, it may not be as adept at supporting weight or volume at this speed. The trend is further accentuated at 0.3 m/s, where even lower ground pressures of 19 kPa to 23 kPa are contrasted with the highest sinkage values, suggesting diminished structural integrity of Bentonite at higher speeds and moisture levels, as illustrated in Table 6.

In contrast, the behavior of diatom soil depicts a different picture. At the base speed of 0.1 m/s, even as ground pressures are notably higher than Bentonite (29 kPa to 33 kPa), sinkage values are comparatively subdued, ranging between 0.8 cm and 1.3 cm. Despite higher ground pressures, the relative stability of Diatom’s sinkage alludes to its potentially higher shear strength or internal friction, possibly attributed to its structural composition. As speeds increase to 0.2 m/s and 0.3 m/s, the ground pressures reduce across the moisture gradient, but there is a more erratic behavior in sinkage values. This erraticism could potentially underscore a complexity in ‘Diatom’s response to mechanical stress, revealing an intricate interplay between its physical structure and moisture content. This could indicate the complex interplay between soil particle arrangement, moisture, and external pressure in Diatom soil, making it react differently than Bentonite.

### 3.4. Impact of Moist Soil Content on Sinkage Exponent

Moisture content significantly affects soil mechanical properties, especially sinkage. Higher moisture content correlates with an increased sinkage exponent, indicating reduced load-bearing capacity as saturation levels rise. Elevated moisture content reduces effective stress, diminishing shear strength and increasing deformation. Assessing load-bearing capacity and sinkage propensity must account for soil moisture content. The moisture content and cohesive modulus indicate the bonding force or strength between soil particles, such as sediment or clay, in cohesive soil. It measures the capacity of cohesive soils to withstand applied stress-induced deformation or structural failure. The cohesive modulus is an essential parameter in soil mechanics and geotechnical engineering, as it offers valuable information regarding the mechanical characteristics of cohesive soils and how they react to external loads. The moisture content, cohesive modulus, and sinkage exponent for Bentonite and Diatom soil types are displayed in Figure 13, and comprehensive data are shown in Table 7.

For Bentonite, there is an evident progressive increase in the sinkage exponent with increasing moisture content. Starting at 5% moisture, the sinkage exponent is recorded at 0.3 and ascends steadily, reaching 1.3 at a moisture content of 30%. This indicates a direct relationship between moisture levels and the sinkage behavior of Bentonite, suggesting that as the soil becomes wetter, its propensity to sink under pressure magnifies. Given these observations, it can be inferred that the complex interplay between water molecules and Bentonite soil particles may increase flexibility, thereby facilitating more significant sinkage under applied loads. Such behavior can have important implications, especially in construction or agricultural settings where precise knowledge of the soil’s response to moisture is paramount [54].

Contrarily, the cohesive modulus of Bentonite depicts an inverse relationship with moisture. Commencing at a robust 150 kPa at 5% moisture, this value dwindles consistently to 12.5 kPa at 30% moisture. This sharp decline underscores that as Bentonite becomes more saturated, its cohesive strength—or its ability to stick together—diminishes considerably. This weakening of cohesion with increased moisture content aligns with prior observations made in the field, emphasizing water’s critical role in altering soil’s mechanical properties. It is interesting to note that even a slight increase in moisture can lead to significant changes in the cohesive modulus, potentially highlighting the sensitivity of Bentonite to water content.

On the other hand, Diatom soil demonstrates a pattern somewhat parallel to Bentonite but with certain variations. The sinkage exponent for Diatom begins at a lower value of 0.1 for 5% moisture but experiences a consistent surge, reaching 1.1 at 30% moisture. This mirrors the trend observed in Bentonite, pointing to an increased sinkage susceptibility with moisture saturation. The cohesive modulus of Diatom, starting at 75 kPa at 5% moisture, follows a decreasing trajectory similar to that of Bentonite. However, when we reach a moisture content of 30%, the cohesive modulus descends to a mere 6.25 kPa, suggesting that Diatom, at higher moisture levels, may possess even lesser cohesive strength than Bentonite. The behavior examined in Diatom soil further establishes the pivotal role of moisture in dictating the structural characteristics of different soil types. Compared to Bentonite, the steeper decline in the cohesive modulus of Diatom might indicate the inherent differences in their compositions and how they interact with water.

The vehicle Is equipped with high-precision pressure sensors strategically placed at contact points between the tracks and the soil. A 6-channel FD0843 data acquisition system (illustrated in Appendix A), along with Computer-Enabled Ground Pressure Measurement User-written Wface, Ver 1.6 Software (elucidated in Appendix A), records real-time pressure sinkage data at varying moisture levels and different sinkage in centimeters, as explained in Figure 12. A detailed analysis and interpretation are presented in Table 7. The results of the rubber-tracked vehicle testing revealed that the pressure sensor readings increased as soil moisture content and vehicle speed increased, as illustrated in Table 6. Compared to diatom soil, the results from the pressure sensor demonstrated that bentonite soil had consistently higher values. This is the explanation for the fact that bentonite soil contains a greater quantity of clay than diatom soil.

When compared to sand particles, clay particles have a higher porosity and smaller diameters than sand particles. It would imply that they can retain more water and produce a more closely linked soil than other soils. Because of the higher clay concentration in bentonite soil, vehicles have a more significant hurdle to penetrate, resulting in increased pressure sensor readings. Table 5 displays the recorded pressure sensor readings from the data logger during vehicle track testing. The experiment examined a soil bin composed of a mixture of Bentonite/Diatom soil and sand and gravel particles ranging from 2–5 mm. The moisture levels of the soil bed were set at 10%, 20%, and 30%, while the vehicle’s speed was set at 0.1, 0.2, and 0.3 m/s.

Figure 12 depicts pressure changes measured by six sensors positioned in the soil bin at different moisture levels of Bentonite/Diatom (10%, 20%, and 30%) and various speeds of the rubber-tracked vehicle (0.1, 0.2, and 0.3 m/s) as it traverses the terrain. The corresponding graph presents tabulated data indicating the pressure measurements recorded by each sensor (Sensor 1 to Sensor 6) at varying speeds and moisture levels. Detailed data are shown in Table 6. This thorough presentation facilitates a meticulous examination regarding pressure distribution among sensors in response to changes in moisture content and vehicle speed. The plotted data provide vital information about the dynamic interactions between the rubber-tracked vehicle and the soil, allowing for a sophisticated comprehension of sinkage pressure patterns under specified experimental conditions.

The graph explains the complex correlation between pressure, moisture levels, and speed throughout all the sensors installed in the soil bin. The X-axis represents moisture levels ranging from 10% to 30%. Simultaneously, the y-axis represents pressure measurements in kilopascals (kPa). Each line on the graph represents a distinct speed category (0.1 m/s, 0.2 m/s, 0.3 m/s), and the individual data points indicate the pressure measurements at specific moisture levels for each of the six sensors. The visual depiction directly compares pressure sinkage data for each sensor in various situations. The graphical illustration enhances comprehension of the intricate trends and patterns in the dataset, offering vital insights into the dynamics of pressure fluctuations concerning moisture levels and vehicle speeds (Figure 14, Table 8).

The pace of the vehicle also significantly influenced the measurements of the pressure sensor. The pressure sensor data exhibited a positive correlation with the vehicle’s speed. At higher speeds, the vehicle exerts tremendous stress on the terrain. A more potent force compacts the soil considerably, leading to elevated readings on the pressure sensor.

The study results show the potential implications for the development of off-road vehicles. The data gathered through laboratory-controlled experiments indicate that the soil type and moisture content can significantly impact the readings from the pressure sensor. These data can be used to design vehicles better adapted for various terrain types. The findings also suggest that the vehicle’s speed can significantly impact the pressure sensor measurements. This information can be used to design vehicles that are optimally adapted for various operating velocities. Overall, the study, which encompasses track vehicle analysis, sinkage observations, soil behavior, ground pressure, and sinkage tests, enhanced our understanding of track vehicles’ interaction with the soil. These insights have practical applications in off-road vehicle design, soil compaction assessment, and predicting soil behavior under varying loads.

## 4. Conclusions

This study comprehensively evaluated ground pressure, bearing capacity, and sinkage in rigid-flexible tracked vehicles operating on characterized terrain under laboratory conditions. Significant insights into pressure sinkage dynamics were gained through experimentation with Bentonite and Diatom sedimental soils and analyzing a rubber-tracked vehicle’s performance. The experimental procedure encompassed Cone Penetration Testing (CPT) and meticulous soil sampling, ensuring a robust foundation for the study. Mathematical relationships for pressure-sinkage and compaction resistance were extensively investigated using established models such as Bekker’s and Rowland’s formulations, shedding light on the intricate relationships between soil properties and vehicle behavior. Furthermore, integrating a multi-body dynamic simulation model employing RecurDyn simulation techniques provided valuable predictive capabilities. It yielded insightful results regarding the interaction between the tracked vehicle and the terrain. Overall, this research contributes significantly to understanding vehicle-terrain interactions and offers valuable implications for the design and operation of tracked vehicles in various soil conditions.

An innovative user-written subroutine for characterizing the unique characteristics of the soil bed sediment has been developed and linked into the RecurDyn simulation environment, based on which a new 3D multi-body simulation model of the tracked vehicle has been developed. Simulations under various conditions prove that the tracked vehicle has good geometric trafficability, bearing trafficability, and locomotion performance in the soil bin. This study validated that sinkage prediction can be enhanced by considering both the pressure-sinkage and shear stress-slip displacement relationships rather than relying solely on the pressure-sinkage relationship. This approach allows for more precise forecasts of sinkage and motion resistance, addressing the tendency for underestimation.

This model reveals a unique property. The traction force initially increases from 0.1 m/s to 0.3 m/s, displaying the maximum traction force at the moist sedimental terrain and constantly recorded values. The noticeable variations in the sinking behavior found between Bentonite and Diatom soils, especially under different moisture levels, indicate the multifaceted nature of this phenomenon. As the traction force decreases at slips higher than the optimum slip, a lack of traction force or serious sinkage will be more likely on seafloor sediments. By applying Terramechanics principles and utilizing advanced sensor technologies such as CPT, moisture sensors, and data loggers, the study has contributed to a comprehensive understanding of the dynamic interaction between rubber-track vehicles and the soil substrate. The investigation conducted in a controlled soil bin environment, utilizing a composite soil mixture of Bentonite/Diatom and sand gravel mixture under moist conditions, has allowed for the precise analysis of pressure-sinkage relationships and bearing capacity performance.

## Figures and Tables

**Figure 1 sensors-24-01779-f001:**
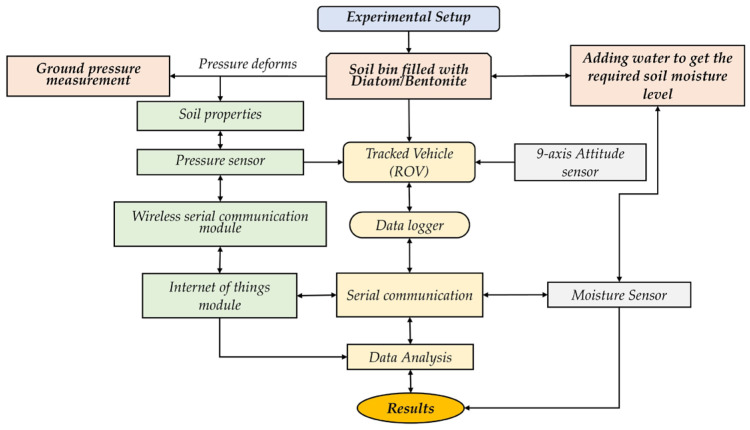
Flow diagram of complete ground instrumentation system for pressure sinkage measurement experiment.

**Figure 2 sensors-24-01779-f002:**
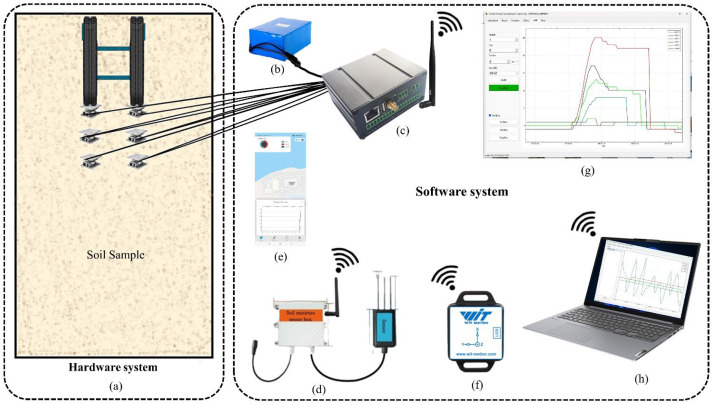
Model experiment schematic diagram includes (**a**) soil bin, pressure sensors, tracked vehicle, (**b**) 12 V/24 V power supply, (**c**) wireless data logger and signal amplifier, (**d**) moisture sensor, (**e**) online monitoring mobile app layout (**f**) 9-axis attitude sensor, (**g**) software, and (**h**) communication module.

**Figure 3 sensors-24-01779-f003:**
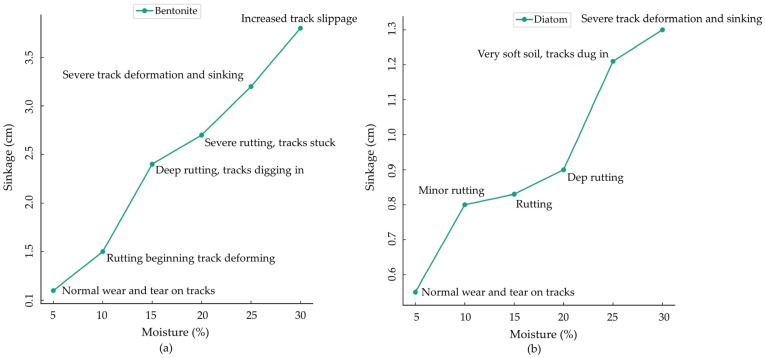
(**a**) Bentonite and (**b**) Diatom soil rubber tracked vehicle pressure sinkage observation at 0.1 m/s speed.

**Figure 4 sensors-24-01779-f004:**
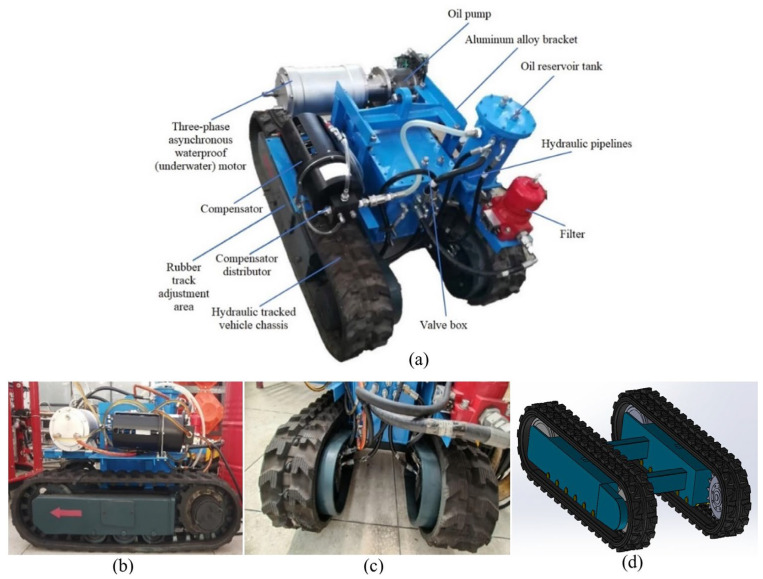
(**a**) Image of bodies of a mini rigid tracked vehicle; (**b**) Proposed tracked vehicle’s side view; (**c**) Proposed tracked vehicle’s front view; (**d**) Solid works schematic track diagram.

**Figure 5 sensors-24-01779-f005:**
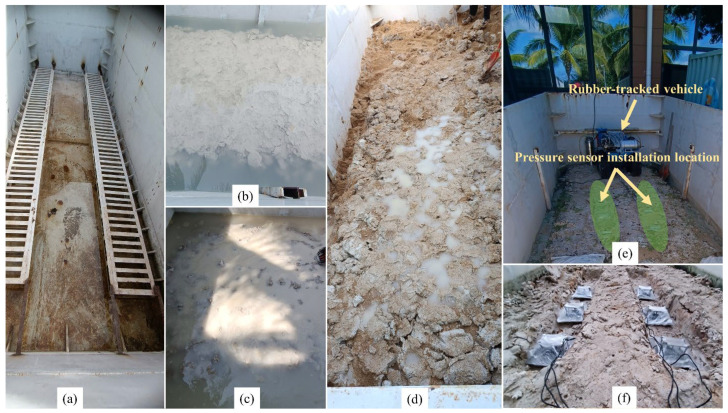
Experimental setup for Soil Bin and pressure sensor installation location: (**a**) Soil Bin used for the experiment, (**b**) Diatom soil in soil bin, (**c**) Bentonite soil in soil bin, (**d**) Sand gravel mixture mixed with bentonite soil, (**e**) Tracked vehicle and pressure sensor installation location, (**f**) front, mid, and Rear pressure sensor location in soil bin.

**Figure 6 sensors-24-01779-f006:**
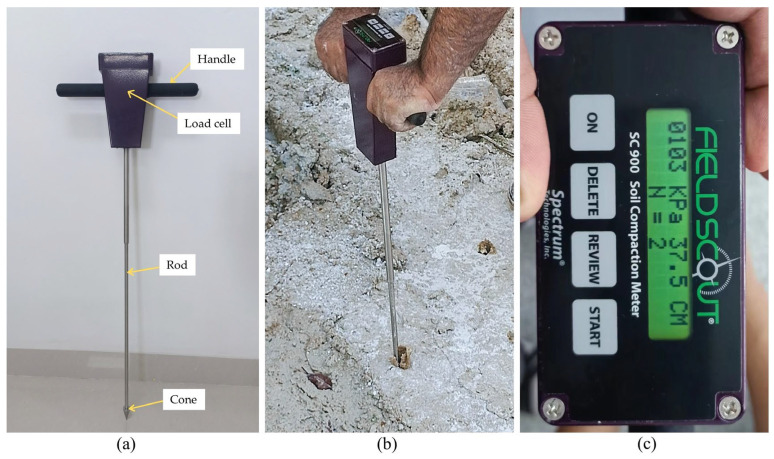
(**a**) Cone penetrometer, (**b**) Soil sampling by cone penetrometer, and (**c**) CPT reading.

**Figure 7 sensors-24-01779-f007:**
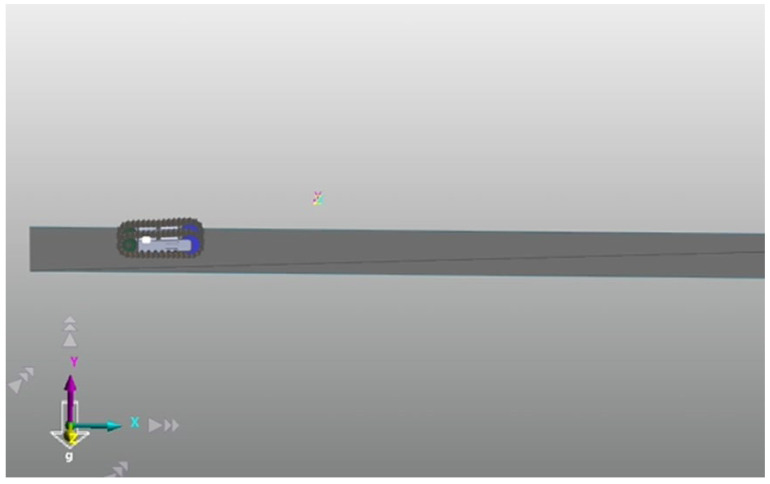
3-D multi-body dynamic simulation model of a tracked vehicle in RecurDyn.

**Figure 8 sensors-24-01779-f008:**
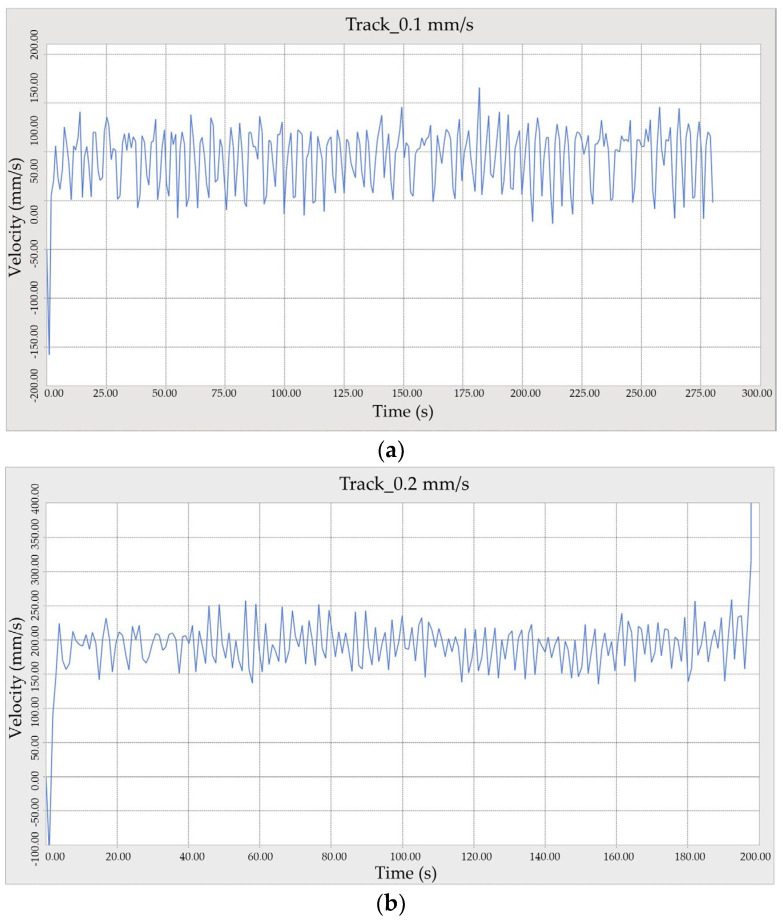
Simulation comparisons: movement of the crawler vehicle on Bentonite when the speed is (**a**) 0.1 m/s, (**b**) 0.2 m/s, (**c**) 0.3 m/s.

**Figure 9 sensors-24-01779-f009:**
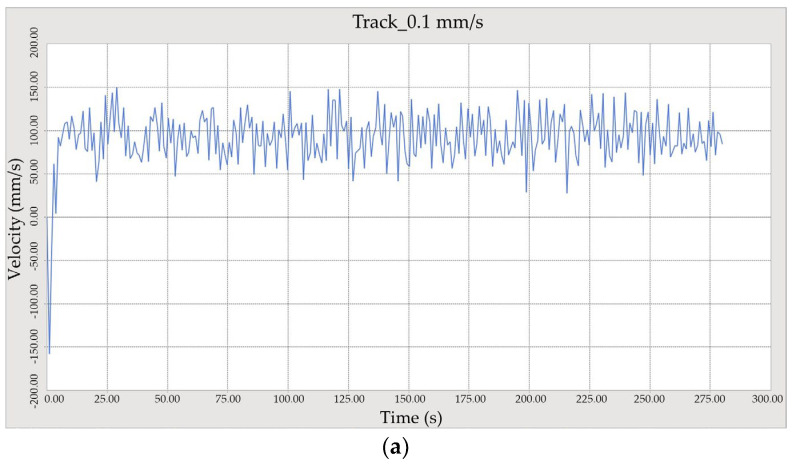
The movement of the crawler vehicle on Diatom when the speed is (**a**) 0.1 m/s, (**b**) 0.2 m/s, (**c**) 0.3 m/s.

**Figure 10 sensors-24-01779-f010:**
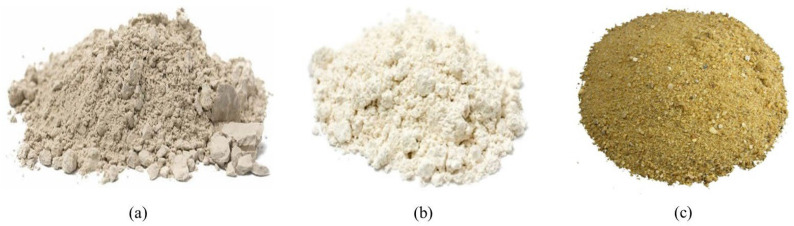
(**a**) Bentonite clay (−0.063 mm), (**b**) Diatom soils (>20 μm), and (**c**) Sand-gravel mixture (2~5 mm) were used in the experiment.

**Figure 11 sensors-24-01779-f011:**
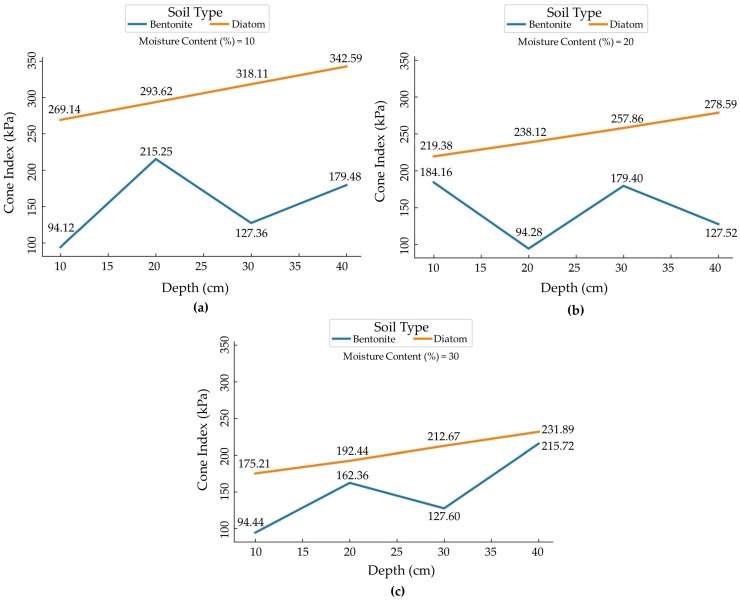
Figure shows the correlation between penetration depth and Cone Index values for Bentonite and Diatom soils at 10%, 20%, and 30% moisture levels. (**a**) 10%, (**b**) 20%, and (**c**) 30% Moisture Content and Penetration Depth.

**Figure 12 sensors-24-01779-f012:**
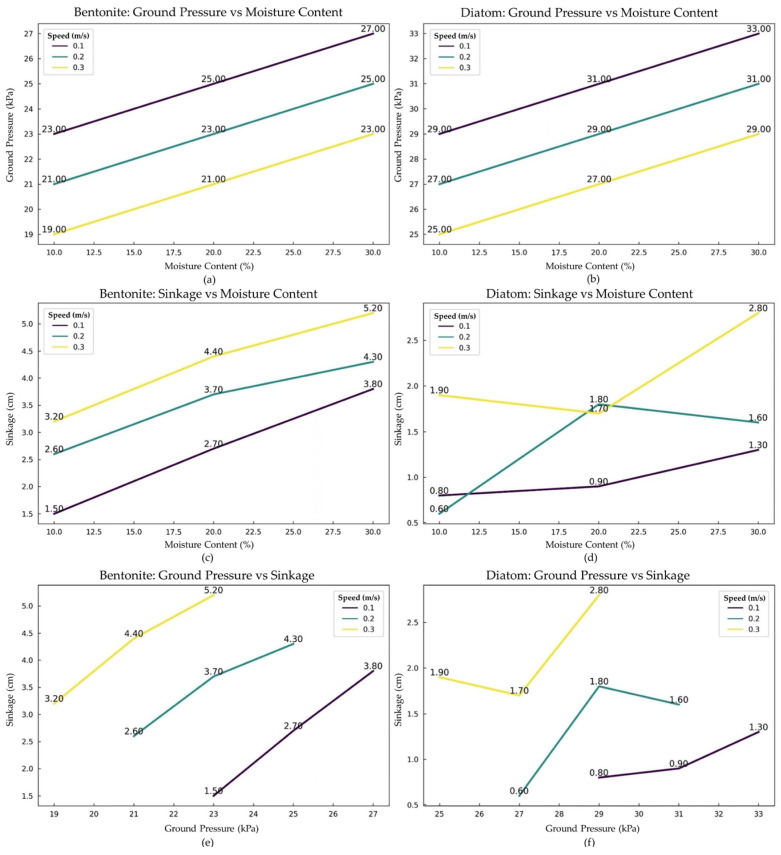
The graph shows (**a**) Bentonite’s moisture (%) on the x-axis and ground pressure (kPa) on the y-axis. (**b**) shows Diatom’s moisture (%) on the x-axis and sinkage (cm) on the y-axis. (**c**) Bentonite’s moisture (%) is on the x-axis, and sinkage (cm) is on the y-axis. (**d**) Shows Diatom’s moisture (%) on the x-axis and sinkage (cm) on the y-axis. (**e**) Bentonite’s ground pressure (kPa) is on the x-axis, and sinkage (cm) is on the y-axis. (**f**) Diatom’s ground pressure (kPa) is on the x-axis, and sinkage (cm) is on the y-axis.

**Figure 13 sensors-24-01779-f013:**
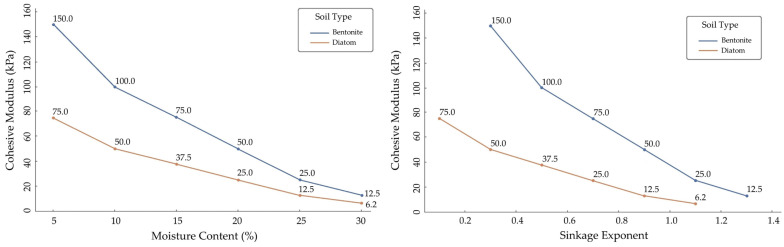
Bentonite and diatom sinkage exponent, moisture, and cohesive modulus.

**Figure 14 sensors-24-01779-f014:**
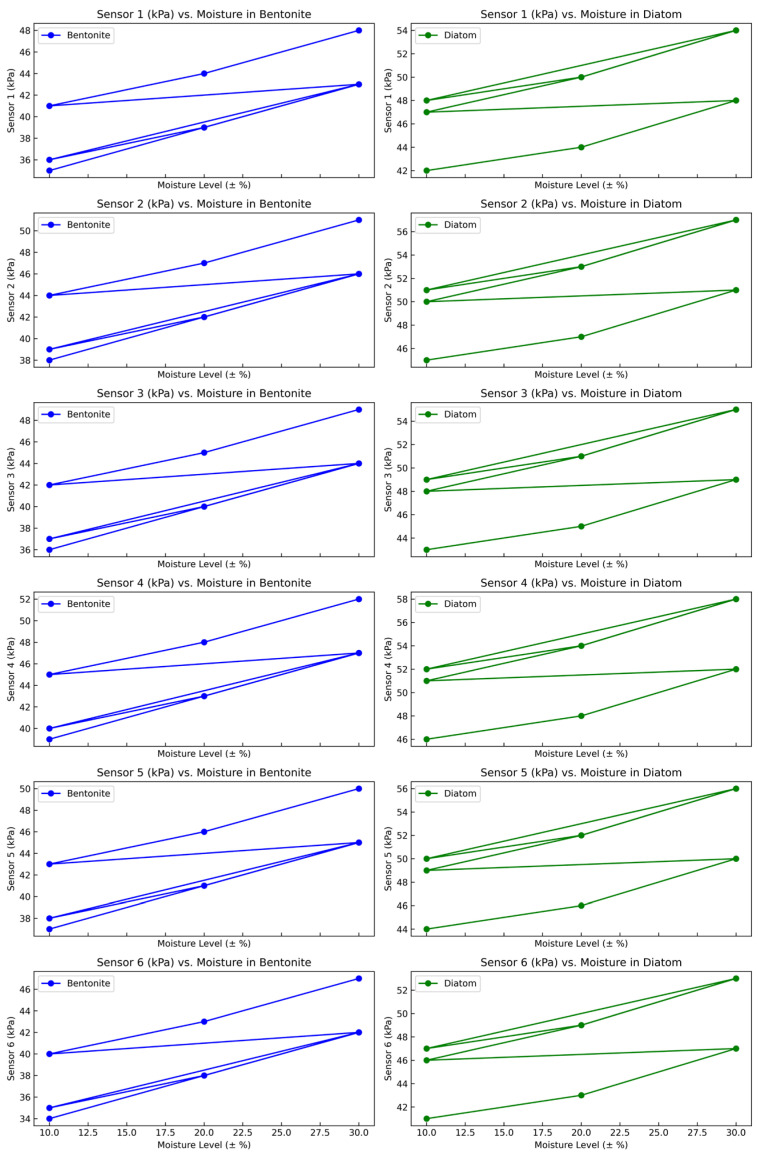
A graphical representation of Sensors Data collected from 6 pressure sensors at variable speeds (0.1, 0.2, and 0.3 m/s).

**Table 1 sensors-24-01779-t001:** Overview of apparatuses used and measurement instrumentations.

Items	Model	Operating Voltage	Measuring Range	Measurement Accuracy
Pressure sensor (Bengbu Zhongnuo Sensor Co., Ltd., Bengbu, China)	ZNHM-D1-2T-22121401	5–15 V DC	214.5 kPa −40 to 85 °C	99.9% ±0.5% FS
Moisture sensor (Weihai JXCT Electronics Co., Ltd., Weihai, China)	JXBS-3001-TR_4G	12–24 V DC	0–100% −40–80 °C	3% in the range of 0–53% 5% in the range of 53–100%
Cone Penetrometer (Spectrum Technologies, Inc., Aurora, CO, USA)	SC 900	4 AAA alkaline batteries	0–18″ (0–45 cm) 0–1000 PSI (0–7000 kPa)	±0.5″ (±1.25 cm) ±15 PSI (±103 kPa)
Data logger built-in Wireless serial communication module (Arizon Company, Shanghai, China)	FD0843 (6 Channel)	9~24 V DC, <10 W	100/200/400/800/1600	Temp: −30~65 °C Relative humidity: 10~95% A/D Update rate: 1600 Hz Data transfer rate:100 Mb/s
Direct Shear Apparatus (Direct Shear Apparatus: GTJ Test Company, Cangzhou, China)	ZJ-1B	110–240 V AC	Shear Displacement: 0–50 mm Normal Load: 0–400 kN	Shear Displacement: ±0.01 mm Normal Load: ±0.5% of applied load Shear Load: ±0.5% of measured value
9-axis attitude sensor (WIT Motion company, Shenzhen, China)	WT901BLECL5.0	3.7 V–260 mAh	TCP: 1~10 Hz UDP: 1~200 Hz	Angular accuracy X, Y axis 0.2°, Z axis 1°

**Table 2 sensors-24-01779-t002:** Vehicle pressure sinkage test observations @ 0.1 m/s velocity.

Soil	Moisture (±%)	Speed (m/s)	Sinkage (cm)	Track-Soil Observations
Bentonite	5	0.1	1.10	Normal wear and tear on tracks
Bentonite	10	0.1	1.50	Rutting beginning tracks deforming
Bentonite	15	0.1	2.40	Deep rutting, tracks digging in
Bentonite	20	0.1	2.70	Severe rutting, tracks stuck
Bentonite	25	0.1	3.20	Severe track deformation and sinking
Bentonite	30	0.1	3.80	Increased track slippage
Diatom	5	0.1	0.55	Normal wear and tear on tracks
Diatom	10	0.1	0.80	Minor rutting
Diatom	15	0.1	0.83	Rutting
Diatom	20	0.1	0.90	Deep Rutting
Diatom	25	0.1	1.21	Very soft soil, tracks dug in
Diatom	30	0.1	1.30	Severe track deformation and sinking

**Table 3 sensors-24-01779-t003:** Basic parameters of the tracked vehicle used for the experiment.

Parameter Name	Symbol	Parameter Content
Track length × width × Height (cm)	Lx × Ly × Lz	120 × 90 × 80
Contact length of rubber track	cm	90
Width of single rubber track	cm	20
Drive wheel diameter	cm	26
Front idler diameter	cm	32
Lugs	cm	19
Weight of the tracked vehicle	W (kg)	544
Weight of the vehicle in the water	W_w_ (kg)	229.25
Contact Pressure	P (kPa)	13.889
Contact Pressure in the water	P_w_ (kPa)	12.5

**Table 4 sensors-24-01779-t004:** Cone penetration resistance at different moistures.

Soil	Depth (cm)	Cone Penetration Resistance kPa @ Moisture (±%)		
		10 ± %	20 ± %	30 ± %
Bentonite	0–5	10 kPa	20 kPa	40 kPa
	5–15	15 kPa	30 kPa	45 kPa
	15–30	20 kPa	25 kPa	50 kPa
Diatom	0–5	15 kPa	18 kPa	20 kPa
	5–15	18 kPa	20 kPa	22 kPa
	15–30	20 kPa	22 kPa	25 kPa

**Table 5 sensors-24-01779-t005:** Rubber-tracked vehicle’s RecurDyn simulation parameters.

Parameter	Unit	Deep Sea Bentonite Sediment	Deep Sea Diatom Sediment
Cohesive deformation modulus	k_c_	7.49	13.85
Internal friction deformation modulus	k_Ø_	0.137	0.23
Deformation index	N	0.34	0.49
Cohesion	C	0.021	0.007
Internal friction angle		16	31
Shearing Deformation modulus	k	8	16
Sinkage Ratio		0.79	0.87

**Table 6 sensors-24-01779-t006:** Results of sinkage, ground pressure, and speed tests for the rubber-tracked vehicle.

Soil Type	Speed (m/s)	Moisture Content (±%)	Ground Pressure (kPa)	Sinkage (cm)
Bentonite	0.1	10	23	1.5
Bentonite	0.1	20	25	2.7
Bentonite	0.1	30	27	3.8
Bentonite	0.2	10	21	2.6
Bentonite	0.2	20	23	3.7
Bentonite	0.2	30	25	4.3
Bentonite	0.3	10	19	3.2
Bentonite	0.3	20	21	4.4
Bentonite	0.3	30	23	5.2
Diatom	0.1	10	29	0.8
Diatom	0.1	20	31	0.9
Diatom	0.1	30	33	1.3
Diatom	0.2	10	27	0.6
Diatom	0.2	20	29	1.8
Diatom	0.2	30	31	1.6
Diatom	0.3	10	25	1.9
Diatom	0.3	20	27	1.7
Diatom	0.3	30	29	2.8

**Table 7 sensors-24-01779-t007:** Moisture content, cohesive modulus, and sinkage exponent of Bentonite and Diatom.

Soil Type	Moisture Content (±%)	Sinkage Exponent	Cohesive Modulus (kPa)
Bentonite	5	0.3	150
Bentonite	10	0.5	100
Bentonite	15	0.7	75
Bentonite	20	0.9	50
Bentonite	25	1.1	25
Bentonite	30	1.3	12.5
Diatom	5	0.1	75
Diatom	10	0.3	50
Diatom	15	0.5	37.5
Diatom	20	0.7	25
Diatom	25	0.9	12.5
Diatom	30	1.1	6.25

**Table 8 sensors-24-01779-t008:** Pressure sensor readings at 10%, 20%, and 30% moisture levels and speeds of 0.1, 0.2, and 0.3 m/s.

Soil Type	Moisture Level (±%)	Speed (m/s)	Sensor 1 (kPa)	Sensor 2 (kPa)	Sensor 3 (kPa)	Sensor 4 (kPa)	Sensor 5 (kPa)	Sensor 6 (kPa)
Bentonite	10	0.1	35	38	36	39	37	34
Bentonite	20	0.1	39	42	40	43	41	38
Bentonite	30	0.1	43	46	44	47	45	42
Diatom	10	0.1	36	39	37	40	38	35
Diatom	20	0.1	39	42	40	43	41	38
Diatom	30	0.1	43	46	44	47	45	42
Bentonite	10	0.2	41	44	42	45	43	40
Bentonite	20	0.2	44	47	45	48	46	43
Bentonite	30	0.2	48	51	49	52	50	47
Diatom	10	0.2	42	45	43	46	44	41
Diatom	20	0.2	44	47	45	48	46	43
Diatom	30	0.2	48	51	49	52	50	47
Bentonite	10	0.3	47	50	48	51	49	46
Bentonite	20	0.3	50	53	51	54	52	49
Bentonite	30	0.3	54	57	55	58	56	53
Diatom	10	0.3	48	51	49	52	50	47
Diatom	20	0.3	50	53	51	54	52	49
Diatom	30	0.3	54	57	55	58	56	53

## Data Availability

New data was investigated in this study. Data sharing does not apply to this article.

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
