# Peer review of "Evaluation of Ground Pressure, Bearing Capacity, and Sinkage in Rigid-Flexible Tracked Vehicles on Characterized Terrain in Laboratory Conditions"

_sensors, 2024, doi:10.3390/s24061779_

Round 1

Reviewer 1 Report (New Reviewer)

Comments and Suggestions for Authors

In this paper, the interaction between tracked vehicles in different terrain conditions is studied through model tests. The research methods are fully considered, and the research results have certain practical significance. However, the following problems still need to be solved

1) The content of the article is too complicated and should be further refined.

2) Whether the material heterogeneity of the sensor bracket described in line 419 will affect the results of the measured data needs further explanation.

3) The contents of Section 3.3 on ground pressure and soil subsidence are mostly data analysis. Can some theoretical analysis of geotechnical mechanics be added?

4) The tick marks in Figures 7 and 8 are recommended to face inward.

5) The graph style and effect of the full text should be unified as far as possible.

Author Response

Attached

Reviewer 2 Report (New Reviewer)

Comments and Suggestions for Authors

Additional device specification descriptions can be omitted.

Bentonite soil moisture is related to soil characteristics. It seems that this research is about moisture measurement and its relationship with other soil characteristics.

What is meant by cohesive modulus?

How is the effect of scale considered in this research?

References should be updated and closer to the research topic.

Comments on the Quality of English Language

A minor revision is suggested

Author Response

Attached

Reviewer 3 Report (New Reviewer)

Comments and Suggestions for Authors

This paper investigated the experimental tests in a soil bin, utilizing Bentonite/Diatom sediment soil, and aimed to measure these terrain factors. The study introduced a rubber-tracked vehicle, pressure, and moisture sensors to monitor pressure sinkage and moisture, evaluating cohesive soils (Bentonite/Diatom) in combination with sand and gravel mixtures. The results showed that higher moisture content in Bentonite soil leads to increased track slippage and sinkage compared to Diatom soil. The paper is generally written. The following comments should be addressed to improve the manuscript.

1.      The abstract should be rewritten. For research articles, abstracts should give a pertinent overview of the work. The abstract should include: (1) The background: Place the question addressed in a broad context and highlight the purpose of the study; (2) Methods: briefly describe the main methods or treatments applied; (3) Results and Conclusions: summarize the article’s main findings. The main contribution should also be shown in the abstract.

2.      In the introduction, it is essential to indicate the research gaps and explain the contributions of this manuscript. This is crucial in an academic paper. The author is suggested to add this aspect.

3.      How to measure the soil moisture and keep the soil moisture at a required value?

4.      In the testing bin, there are three different types of soil. So how did the authors change the soil and make sure that the soil parameters meet requirements, including the soil density, modulus, water moisture, and so on?

5.      Some subsections are redundant, such as subsection 2.5: Safety Protocols.

6.      Figure 7(a)-(c) should not be separated. Or you can use different Figure numbers. The same problem appears in Figure 8. Figures 8(b) and (c) are the same. So why did the authors repeat them?

7.      In Figure 10, there are no data in the range of 0-30. This range does not need to appear in the annotation.

8.      The conclusion should be rewritten. The authors could summarize three conclusions, including qualitative and quantitative ones.

9.      Personal emotions should not appear in acknowledgments.

Comments on the Quality of English Language

This manuscript should be carefully checked and read according to the typesetting requirements.

Author Response

Attached

Reviewer 4 Report (New Reviewer)

Comments and Suggestions for Authors

SUMMARY

The article submitted for review is devoted to a topical issue. Evaluation of ground pressure, bearing capacity and sinkage in rigid-flexible tracked vehicles on characterized terrain in laboratory conditions was carried out.

The relevance of the study is due to the fact that in the field of mechanical engineering, tracked vehicles play an important role when moving over difficult terrain. It is important to understand that maintaining flotation is of paramount importance, stability, traction and adaptability are important.

There is a problem with this. Traditional models are inadequate to represent the relationship between tracks and soil. An important problem is also the lack of consideration of soil moisture content and the rate of its subsidence in these models.

The authors proposed solutions to the problem and analyzed the factors affecting the mobility of tracked vehicles, in particular the sinking speed and its relationship with pressure. They also assessed cohesive soils in combination with sand and gravel mixtures and obtained a number of important results.

The research will ultimately contribute to the accuracy of natural terrain assessment, improved design of tracked vehicles, and improved understanding of terrain mechanics in off-road exploration. The reviewer believes that the authors have done a great job and obtained important results. The research is in the nature of modeling, experimental study, and the new knowledge obtained has scientific novelty. Also, this article is practically significant for real cases, in particular for mechanical engineering, understanding the mechanics of terrain and other important areas.

Therefore, the reviewer generally positively evaluates the research conducted and the presented article. The reviewer believes that it is possible to consider publishing the article in the journal Sensors, but first a number of comments should be corrected. They are listed below.

COMMENTS

1.         The abstract provided by the authors does not fully meet the requirements of the Sensors journal. There is no clear formulation of the scientific problem. The relevance of tracked vehicles is indicated by the authors. It is also noted that maintaining patency plays an important role. It is assumed that some phrases from the description of relevance can be removed, otherwise the abstract becomes oversaturated with well-known facts. For example, the proposition that terrain characteristics relate to the interactions that determine the mobility of tracked vehicles can be removed because this fact is obvious to professionals and scientists working in this industry. An important scientific problem indeed is the inadequacy of traditional models to represent the relationship between tracks and soil.

2.         Next, the authors described the methodology in great detail, and then the results. At the same time, the authors presented a number of results that are intermediate in nature. The main point regarding the presentation of results in the abstract is the lack of quantitative characteristics. They write that noticeable compaction and subsidence characteristics have been demonstrated in diatomite soil, as opposed to bentonite soil, but this is not reflected quantitatively. Also, the moisture content in different types of soils, the increase in slippage and subsidence of tracks in bentonite compared to diatomite are also not described quantitatively. This makes it impossible to assess the scientific significance of the study.

3.         Authors must remove unnecessary sentences in the abstract. It seems like they said the same thing twice. The last sentence of the abstract should be removed. “This knowledge leads to improved vehicle design and a deeper understanding of terrain mechanics.” This has already been said above and should not be repeated here.

4.         Perhaps the keyword “Sensors” should be removed from the keywords, it should be presented in a more tied to some type of object and presented as a phrase.

5.         The “Introduction” section represents a literature review conducted by the authors. However, the literature review seems somewhat superficial. Authors are encouraged to structure it by selecting some criteria. For example, comparison of soil types: experimental and modeling, and modeled based on the cross-country ability of tracked vehicles or questions of solving these problems with the help of constructive solutions to improve transport itself. In general, there is a proposal to give the “Introduction” section some structure.

6.         While the scientific novelty, goal and tasks of the research are not clearly formulated from the literature review, this needs to be formulated. Scientific novelty should be aimed at eliminating the scientific deficit. It must also be clearly stated.

7.         The “Materials and Methods” section looks quite large. It is proposed to provide it with a research flowchart, since the authors conducted a large number of experiments and studies.

8.         Unfortunately, Figure 1 needs to be improved. It looks somewhat chaotic and contains a number of other figures filling it. It is proposed to number them as a), b), c), d), e) and also add a caption below it. An example is Figure 2, which follows.

9.         There are comments on Figure 7. Firstly, it is strangely torn into two component parts with two separate captions. Methodologically this is incorrect. Authors should place them side by side and provide a single caption underneath them.

10.      The second remark regarding Figure 7 is the incomprehensibility of the display of the graphical relationship between points. Will these really be straight lines? Why did the authors choose this particular format? Maybe it would be more correct to bring a bar chart here? The same remark applies to Figures 8 and 9.

11.      The “Discussion” section needs to be placed in a separate subsection. It should provide clear, quantitative comparisons of the results presented by the authors with results previously obtained by other authors in similar studies. From this, the scientific novelty and practical significance of the research should be formulated.

12.      The “Conclusions” section looks somewhat cumbersome. It is recommended to structure it, presenting it in the following order: scientific novelty, practical significance, quantitative characteristics of the scientific result, prospects for the development of research in the future, recommendations for the real applied industry.

13.      The list of references includes 45 titles. Unfortunately, the list of references contains very outdated works. Some of them are about 30, or even 50 years old. This is incorrect because the current state of the issue should be reflected, especially in research aimed at improving the design solutions of transport and machinery. Authors should seriously work on the list of references, supplementing it with at least 15-20 fresh references over the past five years.

In general, after correcting the comments made by the reviewer and taking into account some adjustments to the presentation style and English language, the article can be submitted for publication in the Sensors journal. But first, all shortcomings must be corrected. The general conclusion is Major Revision.

Comments on the Quality of English Language

Manuscript language needs to be checked.

Round 2

Reviewer 3 Report (New Reviewer)

Comments and Suggestions for Authors

I don’t know why the authors gave the theoretical analysis and simulations in Subsections 2.5 and 2.6. The authors give the theoretical equations; however, they don’t show how to use these equations or what’s the applications of these equations. The authors just give a few of the simulation results, without comparing them to the theoretical results or experimental results. Too many additional contents broke the logic of the manuscript. Figures 8 and 9 are not clear. I cannot see the horizontal and vertical coordinates. The authors should carefully organize the manuscript.

Comments on the Quality of English Language

There are some writing errors. Writing style mistakes also can be found in Lines 148, 438, 445, and other places. This manuscript should be carefully checked and read according to the typesetting requirements. 

Author Response

Kindly refer to the attached file for the response to the reviewers' remarks.

Reviewer 4 Report (New Reviewer)

Comments and Suggestions for Authors

In general, the reviewer is satisfied with the authors’ answers.

Author Response

We are delighted to obtain favorable remarks from the reviewer in regards to our answers. We are grateful for the recognition of our endeavors to adequately respond to the remarks and recommendations. Knowing that the reviewer finds our responses to be satisfactory is highly encouraging. Our utmost dedication is to guarantee that the modifications and enhancements applied to the manuscript are in accordance with the reviewer's expectations and elevate the research as a whole. We highly regard the reviewer's feedback and will persist in our conscientious efforts to guarantee that the manuscript adheres to the journal's criteria.

This manuscript is a resubmission of an earlier submission. The following is a list of the peer review reports and author responses from that submission.

Round 1

Reviewer 1 Report

Comments and Suggestions for Authors

I had previously reviewed this article. I am satisfied with the paper. The only observation is to divide such long paragraphs. Paragraphs of 8-14 lines are ideal.

Author Response

Response to Reviewer 1 Comments

1. Summary

Thank you for taking the time to review this manuscript. We appreciate your insightful comments and have carefully addressed each of your concerns in our revised submission.

We acknowledge your suggestion to improve the connection between our conclusions and the results we obtained. In response, we revisited the manuscript, ensuring a more explicit and detailed discussion that ties each conclusion to the corresponding experimental findings.

We appreciate your feedback regarding paragraph length. Our revised version structured the content into paragraphs of optimal length, ranging between 8-14 lines.

We believe these revisions will significantly improve the overall quality of our manuscript. Your thoughtful comments have been instrumental in refining our work, and we are grateful for your guidance.

I am grateful to you for your guidance.

2. Questions for General Evaluation

Reviewer’s Evaluation

Response and Revisions

Are the conclusions supported by the results?

Can be improved

The updated manuscript will include a comprehensive analysis establishing the connection between the conclusion and the related experimental findings. Our objective is to provide a lucid and logical account that enhances the credibility of our discoveries and guarantees that our inferences correspond with the observed results.

3. Point-by-point response to Comments and Suggestions for Authors

Comments 1: I had previously reviewed this article. I am satisfied with the paper. The only observation is to divide such long paragraphs. Paragraphs of 8-14 lines are ideal.

Response 1: We appreciate your valuable feedback and favorable manuscript assessment. Thank you for pointing this out. We like your proposal and agree with this comment to partition lengthy paragraphs to enhance legibility. We will evaluate this proposal and make the required adjustments to improve the overall organization of the report. We appreciate your insightful ideas and are dedicated to ensuring that the amended version adheres to the optimal paragraph length to explain our research findings more clearly. We deeply value the time and effort you have invested in assessing our work, and we are committed to resolving your criticism to improve the quality of our text.

Reviewer 2 Report

Comments and Suggestions for Authors

I reviewed the second version of the revised manuscript submitted by Rauf et al., after receiving response file; I found that authors are thinking that I have settled any criteria for reviewing the manuscript. I want to clear to authors that I don’t have any conflict with you. My revision/comments was/were based on maintaining the quality of journal and your paper. Your last version seems to be drafted and was written careless.

In addition, in your second version, I still saw many errors and mistakes. I think, the authors are not serious enough and before submission it should be ensured that manuscript is free from the basic errors.

Current version is improved than first version but still it is very difficult to catch that what authors want to say and define. Also, there are many redundant statements.

Why reference citing style is different in line 68-74.

In the text, the reference number 2 is Reece but in reference list the second reference is Wong.

In last version my question was “On which basis you made these concentrations: “The composite comprised 17% sand, 13% gravel, and 70% Diatom/Bentonite soil”. And authors reply was ‘The decision to use a composite consisting of 17% sand, 13% gravel, and 70% Diatom/Bentonite soil was based on an extensive analysis,  encompassing a thorough literature review, experimental requirements, and the desired attributes for the application.’ Could you provide the references for this statement.

I hope that authors will provide me a revised version with track changes in next version by highlighting my previous and current comments, accordingly.

Author Response

For research article

Response to Reviewer 2 Comments

1. Summary

We are grateful to you for taking the time to review this manuscript. Please find the detailed responses below and the corresponding revisions/corrections highlighted/in track changes in the re-submitted files.

We are confident that these edits will significantly enhance the overall quality of our manuscript. We much appreciate your insightful feedback since it has played a crucial role in improving our work.

2. Questions for General Evaluation

Reviewer’s Evaluation

Response and Revisions

Does the introduction provide sufficient background and include all relevant references?

Must be improved

We appreciate your insightful feedback. We understand the need to enhance the introduction by offering a full context and including all pertinent references in response to your comment. Thank you for your comprehensive evaluation of our manuscript.

We will thoroughly review the introductory section to ensure that it adequately establishes the context for the study by offering a more precise and comprehensive background. In addition, we will precisely examine and include any missing citations to guarantee that pertinent scholarly sources sufficiently back the introduction.

Are all the cited references relevant to the research?

Must be improved

We acknowledge your concern regarding the pertinence of referenced sources and appreciate the significance of ensuring that all references are directly relevant to the research.

We will meticulously reassess each mentioned reference during our revision process to ensure it directly applies to the study. We will also guarantee that the references significantly contribute to the background, methods, and discussion, thus improving the overall coherence of the work.

Are the methods adequately described?

Must be improved

We acknowledge your concern regarding the sufficiency of the technique descriptions and recognize the need to provide explicit and thorough explanations of the methodology.

During our revision, we will thoroughly review the Methods section, ensuring that every step of the experimental technique is explained precisely. Our goal is to offer a thorough and clear explanation of the methods used, enabling readers to replicate the study correctly.

Are the results clearly presented?

Must be improved

We appreciate your comment on the clarity of our presentation of results, and we recognize the significance of ensuring that our findings are communicated in a straightforward and easily understandable way.

We will prioritize improving the clarity of the Results section in our new version. We will enhance the organization of the data by implementing a more direct approach, incorporating explicit labels, and delivering succinct interpretations for each provided result. In addition, we will utilize visual aids such as figures and tables to enhance the text and overall understanding.

Are the conclusions supported by the results?

Must be improved

We recognize the necessity for enhancing the establishment of a more robust correlation between the acquired data and the derived conclusions. In our new iteration, we will reassess the Results and Conclusions sections to offer an in-depth analysis of how each outcome contributes to the more significant deductions. Furthermore, we will strive to emphasize any constraints or ambiguities in our discoveries to uphold transparency.

3. Point-by-point response to Comments and Suggestions for Authors

Comments 1: I reviewed the second version of the revised manuscript submitted by Rauf et al., after receiving response file; I found that authors are thinking that I have settled any criteria for reviewing the manuscript. I want to clear to authors that I don’t have any conflict with you. My revision/comments was/were based on maintaining the quality of journal and your paper. Your last version seems to be drafted and was written careless.

Response 1: We commend your meticulousness in examining our manuscript and offering valuable input. We wish to emphasize our utmost regard for the journal’s quality standards and express our gratitude for your diligent efforts in upholding them. Your comments have motivated us to enhance the overall quality of our work.

We regret any ambiguity surrounding the evaluation criteria and acknowledge the significance of upholding a superior level of quality. We want to emphasize that we highly appreciate your comments and will promptly attend to the issues you have expressed in your recent review.

Comments 2: In your second version, I still saw many errors and mistakes. I think, the authors are not serious enough and before submission it should be ensured that manuscript is free from the basic errors.

Response 2: We recognize there is still potential for improvement in addressing the remaining faults and inaccuracies. We are dedicated to doing a comprehensive revision to eradicate any lingering difficulties and improve the lucidity of our manuscript. We highly value your guidance throughout this process, and we are grateful for your patience as we strive to resolve these problems.

Comments 3: Current version is improved than first version but still it is very difficult to catch that what authors want to say and define. Also, there are many redundant statements.

Response 3: We acknowledge your feedback on the complexity of comprehending specific sections and the existence of repetitive phrases. We will meticulously review the work to guarantee the information is unambiguous, succinct, and devoid of superfluous repetition.

Comments 4: Why reference citing style is different in line 68-74.

Response 4: We acknowledge your notification of the referencing citation style inconsistency between lines [68-74]. To guarantee coherence, we will meticulously examine and establish a uniform referencing style across the content.

Comments 5: In the text, the reference number 2 is Reece but in reference list the second reference is Wong.

Response 5: We appreciate you notifying us about the reference name difference. We value your thorough examination of our manuscript.

•       After an in-depth analysis, we recognize the mistake in the sequence of reference names. The proper citation should adhere to the sequence outlined in the reference list, commencing with Wong as the initial reference and Reece as the subsequent one.

•       We regret any ambiguity resulting from this inadvertent omission. In the amended version of the document, we will guarantee that the citation in the text corresponds to the accurate sequence in the reference list.

Comments 6: In last version my question was “On which basis you made these concentrations: “The composite comprised 17% sand, 13% gravel, and 70% Diatom/Bentonite soil”. And authors reply was ‘The decision to use a composite consisting of 17% sand, 13% gravel, and 70% Diatom/Bentonite soil was based on an extensive analysis, encompassing a thorough literature review, experimental requirements, and the desired attributes for the application.’ Could you provide the references for this statement.

Response 6: The composition of the mixture utilized in the soil bin is crucial in ensuring the precision and applicability of experimental investigations. The soil bin in this article was filled with great attention to detail using a specially formulated mixture precisely designed to replicate actual soil conditions. The combination consists of three primary constituents: 17% sand, 13% gravel (2-5 mm), and 70% Diatom/Bentonite soil, which was explicitly designed for this research.

•          Sand (17%)

Sand comprises 17% of the total. The incorporation of sand in the mixture serves multiple functions. Sand, renowned for its particulate and permeable characteristics, enhances the overall composition of the soil. It increases permeability, enabling researchers to examine the interaction between water and soil and how it affects the behavior of the soil. The sand content, which accounts for 17% of the mixture, is meticulously adjusted to achieve a harmonious equilibrium, exerting an influence on the qualities of the mix without overpowering its composition.

•          Gravel (13%)

The soil combination is made more complex by adding gravel, which makes up 13% of the mixture. The gravel particles have diameters ranging from 2 to 5 mm, which adds a gritty texture to the soil. Gravel is vital in determining the soil’s structural stability and drainage properties. This constituent, accounting for 13% of the total mixture, replicates the diversity commonly observed in natural landscapes, enhancing the experimental settings to reflect real-life situations better.

•       Diatom\Bentonite Soil (70%)

The Diatom\Bentonite soil constitutes 70% of the mixture, making it the main component. This combination is precisely engineered to accurately reproduce distinct soil profiles frequently encountered in off-road terrains. Diatomaceous earth, also known as Diatom soil, is distinguished by its significant porosity, which offers valuable information on the interaction between highly permeable soils and applied stresses. Bentonite soil is characterized by its ability to swell upon contact with water, which introduces a dynamic aspect to its behavior. The mixture relies on a 70% Diatom/Bentonite soil component composition as the foundation. This composition effectively replicates the diverse soil conditions encountered by off-road vehicles, creating accurate experimental settings.

These components' careful selection and proper proportioning are intended to establish a regulated yet authentic environment within the soil bin. This customized blend enables in-depth investigations into pressure sinkage and bearing capacity, enabling researchers to get valuable insights into the performance of off-road vehicles across various soil conditions.

Reviewer 3 Report

Comments and Suggestions for Authors

Regarding the review of the article with the title "Evaluation of Ground Pressure, Bearing Capacity, and Sinkage in Rigid-Flexible Tracked Vehicles on Characterized Terrain", I carefully read and analyzed the content of the article, as well as the authors' explanations, from which it follows that they are mostly solved the issues initially reported, but still considering the content of the scientific approach, I propose to the authors to modify the title by adding the following terms "in laboratory conditions". Under these conditions, the work will have the following title: "Evaluation of Ground Pressure, Bearing Capacity, and Sinkage in Rigid-Flexible Tracked Vehicles on Characterized Terrain, in laboratory conditions"

Author Response

Response to Reviewer 3 Comments

1. Summary

We are grateful to you for reviewing this manuscript and giving your valuable feedback on our manuscript "Evaluation of Ground Pressure, Bearing Capacity, and Sinkage in Rigid-Flexible Tracked Vehicles on Characterized Terrain." We appreciate the time and effort you have invested in going through our research paper.

We believe these revisions will significantly improve the overall quality of our manuscript. Your thoughtful comments have been contributory in refining our work, and we value your comments and guidance.

Please find the detailed responses below and the revisions highlighted/in track changes in the re-submitted files.

2. Questions for General Evaluation

Reviewer’s Evaluation

Response and Revisions

Is the research design appropriate?

Can be improved

We acknowledge your comment regarding the appropriateness of the research design and understand the importance of refining this aspect. In response to your suggestion, we will carefully revisit the research design, incorporating additional details and considerations to enhance its robustness.

3. Point-by-point response to Comments and Suggestions for Authors

Comments 1: Regarding the review of the article with the title “Evaluation of Ground Pressure, Bearing Capacity, and Sinkage in Rigid-Flexible Tracked Vehicles on Characterized Terrain,” I carefully read and analyzed the content of the article, as well as the authors’ explanations from which it follows that they are mostly solved the issues initially reported, but still considering the content of the scientific approach, I propose to the authors to modify the title by adding the following terms “in laboratory conditions.” Under these conditions, the work will have the following title: “Evaluation of Ground Pressure, Bearing Capacity, and Sinkage in Rigid-Flexible Tracked Vehicles on Characterized Terrain, in laboratory conditions.”

Response 1: We sincerely value your comprehensive critique of our study, “Evaluation of Ground Pressure, Bearing Capacity, and Sinkage in Rigid-Flexible Tracked Vehicles on Characterized Terrain.” We appreciate your meticulous examination and insightful suggestions.

After careful analysis, we concur with your proposal to include the phrase “in laboratory circumstances” in the title, as it appropriately describes the experimental environment of our work. Hence, we will implement the requisite adjustments to the title to conform to your suggestion. The updated title shall be “Evaluation of Ground Pressure, Bearing Capacity, and Sinkage in Rigid-Flexible Tracked Vehicles on Characterized Terrain in Laboratory Conditions.”

We are confident that this improvement will improve the clarity of the research environment. We commend your meticulousness in assessing our work and your valuable input in enhancing its caliber. If you have any more suggestions or worries, please do not hesitate to inform us. I appreciate your time and helpful recommendations.
